# Node2Net: Node-Specific Parameterization for Expressive Graph Representation Learning

## Abstract

Graph Neural Networks (GNNs) have emerged as powerful tools for graph learning. Classical message-passing GNNs enforce permutation equivariance at the node level and permutation invariance at the graph level, but these symmetries constrain expressiveness, limiting them to the discriminative power of the 1-WL test. Recent advances such as Graph Transformers extend GNNs with global attention and positional encodings, yet still rely on shared graph-level parameters. In this work, we revisit the symmetry–expressiveness trade-off through node-specific parameterization, where each node contains a small trainable neural network-an approach we term Node2Net. Unlike existing methods that represent each node with a static embedding vector, Node2Net represents each node with a parametric function capable of modeling nonlinear feature interactions and adaptive transformations. Node2Net breaks 1-WL indistinguishability and can act as universal approximators capable of representing arbitrarily complex node-level transformations. Its computational and memory costs scale linearly with the number of nodes and remain practical on standard benchmarks. As a fundamental node representation method, Node2Net is model- and task-agnostic and does not change the transductive or inductive generalization properties of GNN backbones. Extensive experiments on multiple benchmarks demonstrate that Node2Net consistently improves over node feature learning methods, traditional message-passing GNNs, and recent Graph Transformers.

## 1 Introduction

Graph Neural Networks (GNNs) have become a cornerstone of modern machine learning on relational and structured data, enabling many applications in a wide range of domains including social networks (Hamilton et al., 2017), recommendation systems (Ying et al., 2018), knowledge graphs (Schlichtkrull et al., 2018), and molecular property prediction (Gilmer et al., 2017; Rong et al., 2020). At the core of GNN research lies the topic of node representation that bounds the expressiveness of a graph learning technique. Existing work usually represents a node with an embedding vector, which can be directly from original data features, or generated from graph structural information (e.g., substructures (Bouritsas et al., 2023), subgraphs(Bar-Shalom et al., 2024)) and spectral information (e.g., positional encoding (Rampášek et al., 2022)). Node embeddings can be learned with either unsupervised (Grover & Leskovec, 2016) or supervised approaches (Kipf & Welling, 2017; Veličković et al., 2018; Xu et al., 2019).

Despite their architectural diversity, most existing GNN models can be characterized by a shared parameterization scheme: a small number of global weight matrices shared across all nodes and layers govern how information is aggregated and transformed throughout the model. Consequently, nodes in different structural or semantic contexts are processed with identical transformation functions, which may hinder expressivity and adaptability in heterogeneous or complex graphs (Alon & Yahav, 2021; Oono & Suzuki, 2020). Theoretically such shared-parameter GNNs are limited to the discriminative power of the 1-Weisfeiler–Lehman (1-WL) test, so they cannot distinguish graphs that are 1-WL-indistinguishable (Xu et al., 2019; Morris et al., 2019). As illustrated in Figure 1, to break the expressiveness barrier of 1-WL, several directions have been explored: high-order GNNs incorporating higher-order neighborhoods or higher-dimensional structures (Morris et al., 2019), MPNNs with node IDs, substructure/subgraph-aware GNNs (Bouritsas et al., 2023; Bevilacqua et al., 2022;

You et al., 2021), Graph transformers with global attention and positional encodings(Dwivedi & Bresson, 2021; Ying et al., 2021; Rampášek et al., 2022; Kreuzer et al., 2021).

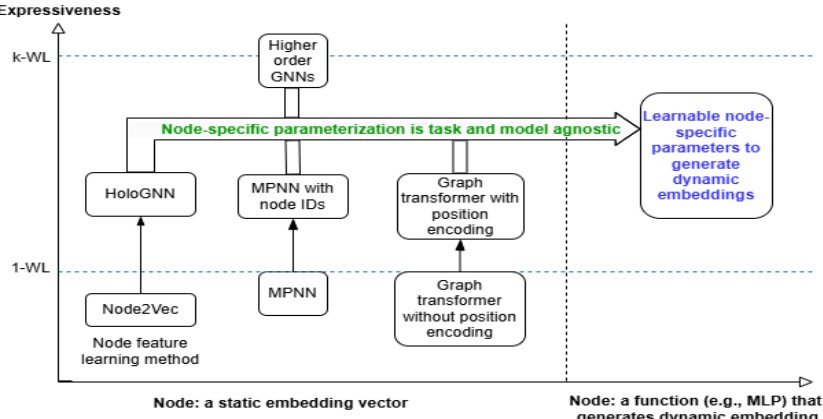

Figure 1: Nodes in existing GNNs (e.g., node feature learning methods, Message passing neural networks, higher-order GNNs, graph transformers) are represented with static embeddings from data or graph structural information. Our approach of node-specific parameterization (Node2Net) represents a node with an expressive function (e.g., MLP) that generates dynamic embeddings.

Among these directions of improving GNN expressiveness, we focus on a fundamental building block for graph modeling: node representation learning. Although adding node IDs (usually random embedding vectors) to GNNs breaks symmetry and provides discriminative power above 1-WL, it faces challenges such as initialization, inductive generalization etc. In this paper we propose *Node-specific Parameterized GNNs* (abbreviated as Node2Net) to augment existing GNNs with trainable node-specific functions. Each node consists of a small neural network (e.g., a two-layer MLP) whose weights are updated during training alongside global GNN parameters. Intuitively, while shared global parameters enforce inductive bias and generalization, introducing a lightweight local parameter module per node can increase a model's capacity to capture fine-grained, node-specific patterns that are otherwise washed out in homogeneous aggregations. Unlike static node embeddings (Grover & Leskovec, 2016; Perozzi et al., 2014; Bevilacqua et al., 2025), Node2Net can model nonlinear feature interactions (e.g., feature interaction in an XOR function), allow identical features at different nodes to map to different outputs, and in principle act as universal approximators capable of representing arbitrarily complex node-level transformations. Node2Net is architecture-agnostic and can be applied to node feature learning methods (Grover & Leskovec, 2016; Bevilacqua et al., 2025), traditional GNN models such as GCN (Kipf & Welling, 2017), GraphSAGE (Hamilton et al., 2017), GAT (Veličković et al., 2018), and recent Graph Transformers (Ying et al., 2021; Rampášek et al., 2022). While this parameterization sacrifices strict permutation invariance, in many practical domains node identity is semantically meaningful and invariance is not required.

In summary, Node2Net introduces a fundamental architectural principle for GNNs, whose applicability extends beyond node classification to encompass link prediction and graph-level learning. Although this paper focuses on parameterizing nodes in a graph, the general idea of component-specific parameterization can be readily extended to edge-centric representations (e.g., in our extension on Graph-GPS in Section 4.3) or tuple-centric representations for k-GNNs discussed in Appendix A. This work thereby delineates a new architectural design space for GNNs and underscores the importance of systematically investigating the interaction between global structural characteristics and localized computations. Our main contributions are:

- **A Novel Node-specific Parameterization Paradigm called Node2Net.** Node2Net introduces node-specific trainable functions for GNNs. Node2Net is versatile, flexible, and model- and task-agnostic, and can be integrated with most GNN models including node representation methods, MPNNs, and graph transformers.
- **Theoretical Insights.** We show that Node2Net is strictly more expressive than static node embeddings, capable of breaking 1-WL indistinguishability.

- **Empirical Validation.** We rigorously evaluate Node2Net by comparing with node representation methods, traditional GNNs, and recent graph transformers using multiple benchmark datasets and demonstrate consistent gains.

## 2 RELATED WORK

Graph learning is a large and active field. Here we focus on work related to node representation and GNN expressiveness. Node representation learns a static embedding vector for each node, e.g., by maximizing the likelihood of random-walk–based neighborhoods (Grover & Leskovec, 2016). More recent holographic node representation targets on generalist node representations capable of solving tasks of any order (Bevilacqua et al., 2025). Spectral methods further advanced this direction (Bruna et al., 2014; Defferrard et al., 2016), culminating in the influential Graph Convolutional Network (GCN) of Kipf & Welling (2017). The message-passing framework was formalized in the Message Passing Neural Network (MPNN) model by Gilmer et al. (2017), encompassing numerous GNN variants. Among these, GraphSAGE (Hamilton et al., 2017) introduced neighborhood sampling to improve scalability, while Graph Attention Networks (GAT) (Veličković et al., 2018) leveraged self-attention for adaptive aggregation. In terms of expressivity, several directions have been explored.

1. **High-order GNNs** are graph neural architectures that extend beyond standard 1-hop message passing by incorporating information from higher-order neighborhoods or higher-dimensional structures (e.g., k-tuples of nodes, subgraphs, or simplicial complexes) to match or exceed the power of k-WL tests, enabling them to capture richer structural patterns, higher-order interactions, and role-based equivalences (Morris et al., 2019). Xu et al. (2019) proposed the Graph Isomorphism Network (GIN), aligning MPNNs with the Weisfeiler-Lehman test. Morris et al. (2019; 2020) extended it with k-GNNs. However, high-order GNNs are often intractable with polynomial complexity. Herbst & Jegelka (2025) extends Invariant Graph Networks to graphons, and shows Invariant Graphon Networks of order k are at least as powerful as the k-WL test.

2. **MPNNs with node IDs** initialize each node with its attributes and a unique identifier embedding, and usually show linear complexity. However, unique node IDs break permutation invariance, and these models are often hard to train, tune, and generalize in practice. Sato et al. (2020) shows nodes assigned with random features can maintain permutation invariance in expectation, but will lose rich feature information.

3. **Substructure/subgraph-aware Substructure-aware GNNs** incorporate graph substructure information (e.g., motifs, walks, paths, or induced subgraphs) into the message-passing process (Bouritsas et al., 2023; Bevilacqua et al., 2022; You et al., 2021). By encoding local or higher-order structural patterns, these methods go beyond standard neighborhood aggregation and achieve greater expressivity than 1-WL, enabling them to capture role similarity, structural equivalence, and higher-order dependencies in graphs.

4. **Graph transformers** extend the Transformer framework by replacing or augmenting the local message-passing mechanism of GNNs with global attention (Dwivedi & Bresson, 2021; Ying et al., 2021; Rampášek et al., 2022; Kreuzer et al., 2021). Unlike traditional MPNNs, which aggregate information only from neighbors, Graph Transformers allow each node to attend to all other nodes (often modulated by positional encodings, structural biases, or sparsity constraints). This enables long-range dependency modeling, higher expressivity beyond 1-WL, and scalability to heterogeneous and large graphs.

Theoretical studies such as Alon & Yahav (2021); Oono & Suzuki (2020); Garg et al. (2020) discusses into expressivity, oversquashing, and generalization bounds, while Du et al. (2019); Poli et al. (2019) explore links between GNNs and spectral or dynamical systems. More recent work studies generalization to arbitary graphs and features with graph foundation models (Finkelshtein et al., 2025), and finer measures of GNN expressiveness (Zhang et al., 2024a; Jin et al., 2024)

## 3 NODE2NET: NODE-SPECIFIC PARAMETERIZED GNNS

GNNs traditionally operate under a paradigm of parameter sharing, where a single set of parameters is globally applied across all nodes and edges in the graph. While this approach offers computa-

tional efficiency and generalizability, it inherently limits the expressiveness of the network, particularly when need to model graphs with node-specific behaviors or heterogeneous structural roles. As shown in Figure 2, Node2Net can be flexibly integrated into a GNN model as long as the model uses an embedding vector to represent a node. Node2Net can take various types of features (e.g., original features from datasets, graph structural features, position encodings, random features) in the input layers, perform complex transformations with the neural network inside a node, and output a new embedding vector of the same size at the output layer. One common issue with node representation methods is that nodes only appearing in testsets are not trained. While we can not solve this issue faced by all node representation methods, we will perform a pre-training step after initializing parameters of all nodes, so the output from each node equals to the input vector before any actual graph learning is conducted. In this way, Node2Net will not affect the transductive or inductive nature of a backbone GNN model, and applies to both types of GNNs.

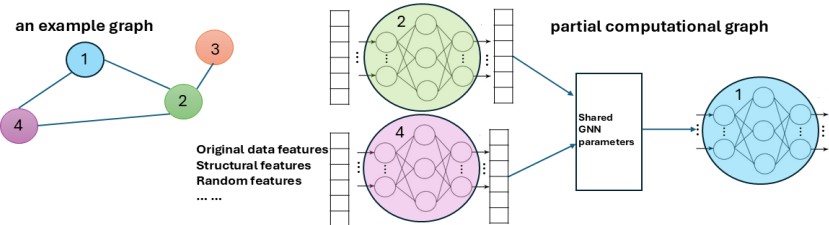

Figure 2: Node2Net can be flexibly integrated into a GNN model's computational graph.

We present how to integrate our Node2Net approach into three popular categories of graph representation models: node representation learning, message passing neural networks (MPNNs), and graph transformers respectively. The case of k-GNNs is discussed in Appendix A.

### 3.1 NODE2NET FOR NODE REPRESENTATION LEARNING

We will use the well-known Node2Vec as a representative of node representation learning methods to show how it can be transformed to Node2Net. Let $G = (V, E)$ be a graph with node features $\{x_v \in \mathbb{R}^{d_{in}} : v \in V\}$. While Node2Vec generates a static embedding vector for each node, Node2Net parameterizes each node with its own lightweight neural function:

$$\Phi_v(\cdot; \theta_v) : \mathbb{R}^{d_{in}} \to \mathbb{R}^d,$$

where $\theta_v$ denotes the trainable parameters of a lightweight nerual network (e.g., a two-layer MLP). The node representation is then

$$h_v = \Phi_v(x_v; \theta_v),$$

with $x_v$ as the input node features (e.g., original attributes, structural features, random features (Abboud et al., 2020), one-hot vector). During training, we generate random-walk contexts $\mathcal{N}_{RW}(v)$ for each node $v$, and maximize the likelihood of context nodes conditioned on $h_v$:

$$\max_{\{\theta_v\}_{v \in V}} \sum_{v \in V} \sum_{u \in \mathcal{N}_{RW}(v)} \log \Pr\left(u \mid h_v\right),$$

where the conditional probability is defined as a softmax

$$\Pr(u \mid h_v) = \frac{\exp(h_u \cdot h_v)}{\sum_{w \in V} \exp(h_w \cdot h_v)}.$$

**Remark: Node2Net generalizes Node2Vec** if each $\Phi_v$ degenerates to a trainable lookup vector (i.e., $\Phi_v(x_v) = e_v$), the formulation reduces to Node2Vec. By using parametric functions, Node2Net can model nonlinear feature transformations, adapt node representations based on input features, and achieve higher representational capacity than static embeddings. Computationally, the number of parameters scales linearly with $|V|$, but $\Phi_v$ can be kept lightweight (e.g., shallow MLPs).

## 3.2 Node-specific parameterized Message Passing Neural Networks

We extend the standard MPNN framework by assigning each node $v \in V$ its own parameterized local function as follows. Let $\theta_v^{(t)}$ denote the trainable parameters of node $v$ at layer $t$. At iteration $t$, the hidden state update is defined as:

$$m_v^{(t)} = \text{AGG}\Big(\{M_v^{(t)}(h_v^{(t-1)}, h_u^{(t-1)}, e_{uv}; \theta_v^{(t)}) : u \in \mathcal{N}(v)\}\Big),$$

$$h_v^{(t)} = U_v^{(t)}\big(h_v^{(t-1)}, m_v^{(t)}; \theta_v^{(t)}\big),$$

where $M_v^{(t)} : \mathbb{R}^d \times \mathbb{R}^d \times \mathbb{R}^{d_e} \to \mathbb{R}^d$ is a message passing function parameterized by $\theta_v^{(t)}$, AGG is a permutation-invariant aggregation (e.g., sum, mean, max), $U_v^{(t)}$ is a update function for node $v$.

After $T$ layers, each node is represented by $\{h_v^{(T)} : v \in V\}$, and each node's evolution depends on its own dedicated parameters. Graph-level outputs can be obtained by applying a readout function:

$$h_G = R\big(\{h_v^{(T)} : v \in V\}\big).$$

This formulation strictly subsumes the standard MPNN: if parameters are tied across all nodes, i.e., $\theta_v^{(t)} = \theta^{(t)}$, we recover the classical shared-parameter MPNN (Gilmer et al., 2017).

## 3.3 Node-Specific parameterized Graph Transformers

With the idea of Node2Net, we extend a graph transformer–style GNN by equipping each node $v \in V$ with its own trainable parameter set $\theta_v$, which defines a lightweight neural network

$$f_{\theta_v} : \mathbb{R}^d \to \mathbb{R}^d,$$

where $f_{\theta_v}$ is the node-specific function whose parameters $\theta_v$ are unique to node $v$. Let $h_v^{(0)} = x_v$ and $h_v^{(t)} \in \mathbb{R}^d$ denote the representation of node $v$ at layer $t$, where $x_v$ is the input feature vector. The attention-based aggregation remains

$$z_v^{(t)} = \sum_{u \in \mathcal{N}(v) \cup \{v\}} \alpha_{vu}^{(t)} W_V^{(t)} h_u^{(t-1)},$$

where $z_v^{(t)}$ denotes aggregated neighborhood message for node $v$ at layer $t$, $\mathcal{N}(v)$ are neighbors of node $v$, $h_u^{(t-1)}$ is the representation of neighbor $u$ from the previous layer, $W_V^{(t)}$ is the global value projection matrix at layer $t$, $\alpha_{vu}^{(t)}$ is attention weight between node $v$ and neighbor $u$.

The attention weights are defined as

$$\alpha_{vu}^{(t)} = \frac{\exp\Big((W_Q^{(t)} h_v^{(t-1)})^\top (W_K^{(t)} h_u^{(t-1)}) / \sqrt{d}\Big)}{\sum_{u' \in \mathcal{N}(v) \cup \{v\}} \exp\Big((W_Q^{(t)} h_v^{(t-1)})^\top (W_K^{(t)} h_{u'}^{(t-1)}) / \sqrt{d}\Big)},$$

The update rule now integrates node-specific transformations:

$$h_v^{(t)} = f_{\theta_v}(U^{(t)}(h_v^{(t-1)}, z_v^{(t)})),$$

where $U^{(t)}$ is the global feed-forward module (e.g., an MLP with residual and normalization layers) shared across all nodes, $f_{\theta_v}$ is node-specific neural network, unique to node $v$, parameterized by $\theta_v$, $h_v^{(t)}$ is the updated representation of node $v$ at layer $t$. Thus, each node learns an individualized parametric mapping that modulates its representation after global self-attention, enabling feature-dependent, node-specific expressiveness beyond uniform parameter sharing.

## 3.4 Theoretical Analysis

A standard *shared-parameter* MPNN computes node states $\{h_v^{(t)}\}$ via globally shared message and update functions. A *Node2Net* model instead associates to each node $v \in V$ a local parametric function $\Phi_v(\cdot; \theta_v)$, such as a small MLP, so that the update rule becomes

$$h_v^{(t)} = \Phi_v\big(U^{(t)}(h_v^{(t-1)}, m_v^{(t)})\big),$$

where $m_v^{(t)}$ denotes the aggregated messages from neighbors and $U^{(t)}$ is a global feed-forward module with residual connections.

**Expressivity beyond 1-WL.** It is well known that shared-parameter MPNNs are upper bounded by the 1-Weisfeiler–Lehman (1-WL) test in distinguishing non-isomorphic graphs (Xu et al., 2019; Morris et al., 2019). Node2Net relaxes this limitation by allowing node-specific mappings.

**Theorem 1** (Node2Net breaks 1-WL indistinguishability)**.** *There exist two non-isomorphic graphs* $G, H$ *such that (1)* $G$ *and* $H$ *are 1-WL indistinguishable, but (2) Node2Net produces distinct node outputs on* $G$ *and* $H$ *for some choice of node-local parameters* $\{\theta_v\}$.

*Proof sketch.* Classical counterexamples (e.g., certain regular graphs) are not separated by 1-WL, hence not by MPNNs. In Node2Net, even if nodes receive identical aggregated inputs, distinct local functions $\Phi_v$ can map these inputs to different outputs, breaking the symmetry. Thus, a parameterization exists that separates $G$ and $H$. $\square$ $\qquad\qquad\qquad\qquad\qquad\qquad\qquad\square$

**Modeling Feature interactions.** Because $\Phi_v$ can be nonlinear (e.g., an MLP with ReLU), Node2Net models higher-order feature interactions. Concretely, for two nodes $u, v$ with identical feature multisets and neighborhoods, a shared-parameter MPNN yields $h_u^{(t)} = h_v^{(t)}$, while Node2Net can produce $h_u^{(t)} \neq h_v^{(t)}$ by using different $\Phi_u, \Phi_v$. This enables the model to distinguish nodes in symmetric roles, which is impossible for 1-WL.

## 3.5 RELATION TO NODE-ID METHODS

A common way to increase GNN expressivity beyond 1-WL is to augment nodes with unique identifiers or random features (Node-ID methods) (Abboud et al., 2020; Sato, 2020; Loukas, 2020). Each node is assigned $v$ an identifier vector $e_v$ (often sampled from a random distribution) and feeds $e_v$ as part of the input feature. This breaks 1-WL indistinguishability, since nodes with identical local neighborhoods can now be separated by their IDs. However, ID-based approaches face two key difficulties (1) **randomness and instability:** Random ID features introduce variance across runs and may require multiple restarts to stabilize performance, and (2) **limited functional role:** IDs act as static tags; they do not provide feature-dependent transformations or model nonlinear interactions between a node's attributes and its structural context. In contrast, Node2Net assigns each node a parametric function $\Phi_v(\cdot; \theta_v)$ instead of a fixed ID vector, which has several advantages:

- **Learnable parameters:** node-local parameters are optimized during training, removing the need for careful stochastic initialization and aligning with the training objective.

- **Beyond tagging:** Whereas node IDs only differentiate nodes by identity, Node2Net enables each node to *transform* its input features and aggregated messages in a node-specific manner. That is, even if two nodes share identical features and neighborhoods, their different $\Phi_v$ mappings can produce distinct outputs.

- **Modeling feature interactions:** Since $\Phi_v$ can be an MLP or other nonlinear module, Node2Net captures nonlinear interactions between input features, node identity, and local context—a capacity entirely absent from pure ID methods.

**Expressivity consequence.** Formally, node-ID augmentation can be seen as the special case of Node2Net where each $\Phi_v$ ignores its input and directly outputs a learnable embedding vector. Thus, Node2Net *strictly subsumes* node-ID methods in representational power: it retains the ability to differentiate nodes by identity while also providing flexible, data-dependent transformations. This additional functional capacity explains why Node2Net can overcome the instability and limited expressivity of random-feature ID methods.

## 4 EXPERIMENTS

We conducted extensive experiments to validate our Node2Net approach by comparing with three categories of GNN methods (node representation methods, traditional GNNs, graph transformers) using 5 graph learning benchmarks (details are given in Appendix B) and two graph tasks (node classification and graph regression).

## 4.1 Experiment with node representation method Node2Vec

For node representation methods, we chose to compare with the classic Node2Vec (Grover & Leskovec, 2016) approach with a 2-phase implementation of Node2Net. In Phase 1, we train Node2Vec to obtain base embeddings. In Phase 2, we insert a lightweight MLP into each node and pretrain this MLP to input and output the same Phase 1 embedding using an $L_2$ reconstruction loss (MSE). We then continue training with skip-gram objective, replacing the embedding lookup with MLP outputs. For each random walk, a *positive sample* is defined as a center node $v$ paired with its context node $u$ from the walk, while *negative samples* are nodes $n_i$ randomly drawn from a noise distribution $P_n$. Following standard setting in Node2Vec, the noise distribution is defined as

$$P_n(v) \propto d_v^{3/4},$$

where $d_v$ denotes the degree of node $v$. The objective encourages large inner products for positive pairs and small inner products for negative pairs:

$$\mathcal{L} = -\Big[ \log \sigma(h_u \cdot h_v) + \sum_{i=1}^{K} \mathbb{E}_{n_i \sim P_n} \log \sigma(-h_{n_i} \cdot h_v) \Big],$$

where $\sigma(\cdot)$ is the sigmoid function.

For evaluation, we follow standard practice in node classification: the learned embeddings are fed into a logistic regression classifier, trained on the training split and evaluated on the test split. We report accuracy (and F1 score for PPI) averaged over 100 random seeds. Table 1 summarizes the performance of two methods. Node2Net consistently outperforms the baseline across datasets, achieving the highest accuracy on Cora and PubMed, and strong improvements in F1 score on PPI.

It is worth noting that we cannot directly use `node2vec.loss()` function provided in PyTorch Geometric, because it only computes losses over an internal embedding lookup table. In our two-phase procedure, embeddings are generated dynamically by node-specific MLPs instead of static lookup vectors. Therefore, we explicitly compute the skip-gram loss with MLP outputs so gradients correctly flow into the MLP parameters. Hence, Node2Net loss values are not directly comparable to Node2Vec losses, but embeddings dynamically generated by Node2Net exhibit improved linear separability, leading to higher downstream classification accuracy as shown in Table 1.

Table 1: Experiment results with node representation method. Reported values are accuracy (%) for Cora, CiteSeer, and PubMed, and Micro-F1 for PPI. More detailed results are in Appendix C.1.

| Method | Cora | CiteSeer | PubMed | PPI (Micro-F1) |
|--------|------|----------|--------|----------------|
| Node2Vec | 68.75 ± 1.18 | 48.63 ± 1.67 | 69.95 ± 0.85 | 0.1911 ± 0.0040 |
| Node2Net | **73.24 ± 0.95** | **51.58 ± 1.15** | **71.97 ± 1.31** | **0.1930 ± 0.0055** |

## 4.2 Experiments with Traditional GNN Models

We chose three widely used traditional GNN models for comparison: GCN (Kipf & Welling, 2017), GraphSAGE (Hamilton et al., 2017), and GATv2 (Veličković et al., 2018). For each model, we construct an enhanced variant by inserting a pretrained MLP (pretrained so initial output equals to input) into each node:

$$X_i' = \begin{cases} \phi_i(X_i), & \text{if } i \in \mathcal{V}_{\text{train}}, \text{then i's MLP is activated}, \\ X_i, & \text{otherwise}. \end{cases}$$

where $V_{\text{train}}$ is the set of nodes included in the training set, each training node $i$ has its own MLP $\phi_i$. The GNN $f_\theta$ then operates on $X'$:

$$Z = f_\theta(X', \hat{A}), \qquad \hat{Y} = \text{softmax}(Z).$$

By construction, validation/test nodes never pass through any Node-MLP, preventing information leakage and keeping inference cost identical to the baseline models.

We adopt a two-phase training procedure to integrate node-specific MLPs into each backbone:

- **Phase 1 (Node MLP pretraining).** Each $\phi_i$ is trained independently on nodes in $\mathcal{V}_{\text{covered}} =$ train nodes $\cup$ their 1-hop neighbors, with reconstruction objective $\phi_i(X_i) \approx X_i$ (MSE). This initializes Node-MLPs to behave like identity mappings.

- **Phase 2 (Joint model training).** The pre-trained Node-MLPs are activated inside the GNN backbone. We then train the full model end-to-end for 200 epochs, allowing node features to evolve dynamically through both message passing and per-node refinement. Unless otherwise specified, backbone parameters remain trainable during this phase. Hyperparameters and detailed experiment settings can be found in Appendix C.5.

As shown in Table 2 and Table 3, Node2Net consistently improves or matches baseline performance:

- **Local parameterization:** Node-specific parameterization improves both message-passing (GCN, GraphSAGE) and attention-based (GATv2) backbones.

- **Robustness:** Gains are stable across citation networks (Cora, CiteSeer, PubMed) and PPI, demonstrating generality. Improvements on loss are often significant.

- **Flexibility:** In featureless graphs (PPI), integrating Node2Net embeddings enables strong performance in both GraphSAGE and GATv2 backbones.

Table 2: Experiment results on (Accuracy / Micro-F1). More detailed results are in Appendix C.2.

| Method | Cora | CiteSeer | PubMed | PPI (Micro-F1) |
|---|---|---|---|---|
| GCN | 83.07 ± 0.79 | 67.89 ± 0.81 | 78.51 ± 0.57 | 0.1749 ± 0.0043 |
| Node2Net-GCN | **83.30 ± 0.73** | **67.95 ± 0.71** | **78.59 ± 0.54** | **0.1770 ± 0.0056** |
| GATv2 | **81.23 ± 1.60** | 69.02 ± 2.00 | 76.91 ± 1.06 | **0.1020 ± 0.0206** |
| Node2Net-GATv2 | 81.01 ± 1.28 | **69.21 ± 1.68** | **77.29 ± 0.71** | 0.1006 ± 0.0201 |
| GraphSAGE | 79.54 ± 0.77 | 70.19 ± 0.55 | 76.61 ± 0.47 | 0.1839 ± 0.0035 |
| Node2Net-GraphSAGE | **79.61 ± 0.85** | **70.48 ± 0.63** | **77.31 ± 0.34** | **0.1848 ± 0.0043** |

Table 3: Experiment results on loss. More detailed results can be found in Appendix C.2.

| Method | Cora | CiteSeer | PubMed | PPI (Loss) |
|---|---|---|---|---|
| GCN | 60.96 ± 2.00 | 116.68 ± 3.70 | 57.63 ± 0.81 | 1.0324 ± 0.0020 |
| Node2Net-GCN | **60.85 ± 2.19** | **116.09 ± 3.28** | **57.57 ± 0.75** | **0.8145 ± 0.0022** |
| GATv2 | 103.88 ± 3.66 | 129.10 ± 1.92 | 61.10 ± 1.76 | **1.2585 ± 0.0289** |
| Node2Net-GATv2 | **87.49 ± 3.24** | **102.86 ± 2.14** | **58.08 ± 1.50** | 1.2592 ± 0.0301 |
| GraphSAGE | **64.87 ± 1.77** | **92.42 ± 0.89** | 62.87 ± 1.05 | 0.6507 ± 0.0038 |
| Node2Net-GraphSAGE | 65.92 ± 2.13 | 92.58 ± 1.23 | **60.66 ± 0.75** | **0.5721 ± 0.0045** |

### 4.3 EXPERIMENT WITH GRAPH TRANSFORMER METHOD GRAPHGPS

For graph transformers, we chose GraphGPS (Rampášek et al., 2022) due to its incorporation of rich graph information. We extend GraphGPS by introducing **NodeEdgeMLP (NE-MLP)** (Appendix §D) to replace static embeddings with categorical MLPs operating on one-hot identifiers for both nodes and edges. This design enhances structural expressivity, incorporates positional encodings, and introduces gradient scheduling to decouple the dynamics of edge and node optimization.

**Baseline GraphGPS initialization:** node and edge features are initialized by embedding lookups:

$$h_i^{(0)} = \left[ \text{Embed}_{\text{node}}(t_i) \parallel W_{\text{PE}} \, \text{PE}_i \right], \qquad e_{ij}^{(0)} = \text{Embed}_{\text{edge}}(r_{ij}),$$

where $t_i$ is the node type, $r_{ij}$ the edge type, and $\text{PE}_i$ the positional encoding.

**NE-MLP initialization (ours).** We replace static embeddings with categorical MLPs:

$$h_i^{(0)} = \left[\, \phi_{\text{node}}\big(\text{onehot}(t_i)\big) \,\|\, \phi_{\text{PE}}(\text{PE}_i) \,\right], \qquad e_{ij}^{(0)} = \phi_{\text{edge}}\big(\text{onehot}(r_{ij})\big),$$

where

$$\phi_{\text{node}} : \mathbb{R}^{|T|} \to \mathbb{R}^{d_h - d_{\text{PE}}}, \quad \phi_{\text{edge}} : \mathbb{R}^{|R|} \to \mathbb{R}^{d_h}, \quad \phi_{\text{PE}} : \mathbb{R}^{d_{\text{PE,in}}} \to \mathbb{R}^{d_{\text{PE}}},$$

with $|T|$ and $|R|$ denoting the number of node and edge types, and $d_h$ the hidden channel size.

**Gradient scheduling.** To reduce optimization noise, we introduce an update mask:

$$\nabla\phi_{\text{node}} \neq 0 \quad \forall\, \text{steps}, \qquad \nabla\phi_{\text{edge}} = \begin{cases} \nabla\phi_{\text{edge}}, & \text{if } s \equiv 0 \pmod{n}, \\ 0, & \text{otherwise}, \end{cases}$$

where $s$ is the training step index and $n$ is the update period (default $n = 5$).

**Discussion and summary of results.**

- **Enhanced structural bias.** NE-MLP maps node types, edge types, and positional encodings into unified channel-aligned features, yielding richer structural encodings compared to fixed embeddings and clear performance improvement as shown in Table 4.

- **Controlled gradient scheduling.** Node MLPs update every iteration while edge MLPs update periodically ($n = 5$ by default), providing smoother training for edge embeddings.

- **Inductive generalization.** No pretraining or auxiliary supervision is applied. Both GPS and Node2Net-GPS are trained from scratch, showing that NE-MLP generalizes without external knowledge transfer.

Table 4: Experiment results on ZINC. More detailed results can be found in Appendix C.3.

| Model | Test mean loss $\pm$ standard deviation |
|---|---|
| GPS | $0.08708 \pm 0.006$ |
| Node2Net-GPS (NE-MLP) | $\mathbf{0.08621 \pm 0.004}$ |

Recent trends in graph learning increasingly emphasize the integration of richer structural, topological, and semantic information from graphs (Hussain et al., 2024; Gao et al., 2024; Zhao et al., 2025), and integration of specially designed architectural component such as Node2Net is often not straightforward and deserves more study. Specifically with graph transformers, their performance is significantly impacted by tokenization (Zhang et al., 2024b; Müller & Morris, 2024), which is central to our Node2Net approach and will be explored in future work.

## 5 CONCLUSION

In this work, we introduced a novel node-specific parameterization method called Node2Net as a principled approach to enhance the expressiveness of GNNs by equipping each node with a learnable function capable of modeling nonlinear feature interactions and feature-dependent variability. This mechanism strictly extends the representational power beyond traditional node embeddings and shared-parameter GNNs, allowing the model to break 1-WL indistinguishability while maintaining linear computational and memory scaling. Importantly, Node2Net does not alter the inherent transductive or inductive generalization properties of the backbone, ensuring applicability to both settings. We demonstrated how this concept can be seamlessly integrated into many widely-used GNN architectures. Empirical results on multiple node classification benchmarks confirm consistent performance gains over node representation methods, classical GNNs, and graph transformers. Node2Net constitutes a fundamental design principle with potential applications beyond node classification, including link prediction and graph-level tasks. Additionally, the idea of component-specific parameterization can also be applied to more graph components, which will be explored in future work. We believe this work opens a new avenue for GNN design and encourages further exploration into the interplay between global structure and local computation.

**Reproducibility statement** We are committed to ensuring the reproducibility of our results. To this end, upon acceptance we will provide:

1. Code and Implementation Details: Our full implementation, including training and evaluation scripts, is available in an open-source repository (link to be released upon acceptance). The code specifies all model architectures, hyperparameters, and random seeds.

2. Datasets: All datasets used in this work (Cora, Citeseer, Pubmed, and others) are publicly available. We include preprocessing scripts to reproduce the exact input data splits used in our experiments.

3. Experimental Setup: We document the computing environment (hardware, software versions, GPU/CPU specifications) and report training times and memory usage.

4. Hyperparameters: All hyperparameters are reported in the Appendix, including learning rates, batch sizes, optimizer settings, regularization coefficients, and early stopping criteria.

5. Statistical Rigor: For each benchmark, results are averaged across multiple runs with different random seeds, and we report both mean and standard deviation. Statistical significance is assessed using paired tests where appropriate.

6. Limitations: While our experiments cover widely used benchmarks, large-scale industrial graphs and certain application domains (e.g., temporal or dynamic graphs) are beyond the scope of this study. Future work will address scalability and broader applicability.

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

## A    NODE2NET FOR HIGHER ORDER K-GNNS

Let $G = (V, E)$ be a graph. For a positive integer $k$, denote by $\mathcal{T}_k = \{S = (v_1, \ldots, v_k) : v_i \in V\}$ the set of ordered or unordered $k$-tuples (we use ordered tuples for notational simplicity). Let $x_v \in \mathbb{R}^{d_x}$ be the input feature of node $v$. For a tuple $S = (v_1, \ldots, v_k) \in \mathcal{T}_k$ we write $x_S$ for a tuple-level feature (e.g., concatenation or a structural feature):

$$x_S = \text{concat}(x_{v_1}, \ldots, x_{v_k}) \in \mathbb{R}^{kd_x}.$$

Here $x_S$ is the input to the $k$-tuple representation function.

### A.1    DIRECT PER-$k$-TUPLE NODE2NET

Assign each $k$-tuple $S \in \mathcal{T}_k$ its own lightweight parametric function $f_{\theta_S} : \mathbb{R}^D \to \mathbb{R}^D$ (where $D$ is the tuple representation dimension). Let $h_S^{(t)} \in \mathbb{R}^D$ denote the representation of tuple $S$ at layer $t$. The high-order attention aggregation and update follow the same pattern as node-level, but over tuple neighborhoods $\mathcal{N}_k(S)$ (tuples adjacent to $S$ under the chosen k-GNN adjacency relation).

$$z_S^{(t)} = \sum_{T \in \mathcal{N}_k(S) \cup \{S\}} \alpha_{S,T}^{(t)} W_V^{(t)} h_T^{(t-1)},$$

$$\alpha_{S,T}^{(t)} = \frac{\exp\big((W_Q^{(t)} h_S^{(t-1)})^\top (W_K^{(t)} h_T^{(t-1)})/\sqrt{D}\big)}{\sum_{T' \in \mathcal{N}_k(S) \cup \{S\}} \exp\big((W_Q^{(t)} h_S^{(t-1)})^\top (W_K^{(t)} h_{T'}^{(t-1)})/\sqrt{D}\big)},$$

$$h_S^{(t)} = f_{\theta_S}\big(U^{(t)}(h_S^{(t-1)}, z_S^{(t)})\big),$$

This is a direct extension: each $k$-tuple has a unique parametric mapping $f_{\theta_S}$. It is expressive but scales as $|\mathcal{T}_k|$, which is typically intractable for moderate-sized graphs.

### A.2    PRACTICAL, PARAMETER-EFFICIENT IMPLEMENTATIONS

Below are two parameter-efficient ways to capture tuple-specific adaptation while avoiding an explosion in parameters.

#### A.2.1    HYPERNETWORK (GENERATE TUPLE PARAMETERS FROM NODE-LEVEL CODES)

Equip each node $v$ with a small code $c_v \in \mathbb{R}^p$ (or node-specific parameters $\theta_v$). Form a tuple code $c_S$ by pooling the node codes:

$$c_S = \text{Pool}(c_{v_1}, \ldots, c_{v_k}),$$

and use a hypernetwork $G_\phi$ to generate lightweight parameters for tuple $S$:

$$\tilde{\theta}_S = G_\phi(c_S).$$

The tuple update becomes

$$h_S^{(t)} = f_{\tilde{\theta}_S}\big(U^{(t)}(h_S^{(t-1)}, z_S^{(t)})\big),$$

where $f_{\tilde{\theta}_S}$ is a small MLP whose parameters are the output $\tilde{\theta}_S$ of the hypernetwork. Only $\{c_v\}_{v \in V}$ and $\phi$ are learned (plus global projection matrices), keeping parameter count manageable. The hypernetwork compresses per-tuple variation into a function of node-level codes. Complexity scales with $|V|$ rather than $|\mathcal{T}_k|$.

#### A.2.2    COMPOSITIONAL PER-NODE FUNCTIONS + FUSION (ELEMENTWISE COMPOSITION)

Instead of generating tuple parameters, apply node-specific transformations to each element in the tuple and then fuse the transformed element representations.

Assign each node $v$ a node-specific function $g_{\theta_v} : \mathbb{R}^{d_x} \to \mathbb{R}^{d'}$. Compute elementwise transformed features and combine:

$$u_i^{(t)} = g_{\theta_{v_i}}\big(\pi_i\big(U^{(t)}(h_S^{(t-1)}, z_S^{(t)})\big)\big), \qquad i = 1, \ldots, k,$$
$$h_S^{(t)} = \mathcal{F}\big(u_1^{(t)}, \ldots, u_k^{(t)}\big),$$

Alternatively, one can use multi-head cross-attention among the transformed elements:

$$h_S^{(t)} = \mathrm{CrossAtt}\big([u_1^{(t)}, \ldots, u_k^{(t)}]\big),$$

This scheme only stores per-node parameters $\theta_v$ (as in Node2Net) and a small global fusion network; it is computationally efficient and naturally generalizes standard $k$-GNN architectures.

### A.3 CHOICE OF TUPLE NEIGHBORHOOD AND COMPLEXITY

**Tuple neighborhood** $\mathcal{N}_k(S)$**.** A standard choice: $\mathcal{N}_k(S)$ contains tuples obtained from $S$ by replacing one element $v_i$ with a neighbor $u \in \mathcal{N}(v_i)$. Formally, for ordered tuples:

$$\mathcal{N}_k(S) = \big\{(v_1, \ldots, v_{i-1}, u, v_{i+1}, \ldots, v_k) : i \in [k], u \in \mathcal{N}(v_i)\big\}.$$

**Complexity observations.**

- Direct per-tuple parameters: memory $\mathcal{O}(|\mathcal{T}_k| \cdot P)$ for param size $P$ — infeasible for large graphs.
- Hypernetwork: memory $\mathcal{O}(|V| \cdot p + |\phi|)$ — scalable when $p \ll P$.
- Compositional scheme: memory $\mathcal{O}(|V| \cdot P_v + P_{\text{fusion}})$ where $P_v$ is per-node MLP size — typically feasible.

## B DATASETS DESCRIPTION

- **Cora:** A citation network with 2,708 nodes and 5,429 edges, where each node corresponds to a scientific publication and edges represent citation links. Each node is assigned to one of 7 classes. Following the GCN paper, we use 20 nodes per class for training (140 in total), 500 nodes for validation, and 1,000 nodes for testing.
- **CiteSeer:** A citation network containing 3,327 nodes and 4,732 edges, categorized into 6 classes. The split uses 20 nodes per class for training (120 in total), 500 nodes for validation, and 1,000 nodes for testing.
- **PubMed:** A large-scale biomedical citation network with 19,717 nodes and 44,338 edges, divided into 3 classes. The split uses 20 nodes per class for training (60 in total), 500 nodes for validation, and 1,000 nodes for testing.
- **PPI:** A subgraph of the Protein–Protein Interaction (PPI) network for *Homo Sapiens*. This subgraph is induced by proteins for which labels are available from hallmark gene sets, representing different biological states. It contains 3,890 nodes, 76,584 edges, and 50 distinct labels, and is evaluated as a multi-label node classification task.
- **ZINC:** A molecular graph regression dataset widely used for benchmarking graph transformers. Each molecule is represented as a graph with atoms as nodes and bonds as edges, and the task is to predict constrained solubility values. We follow example setting in PyG GraphGPS to adopt the standard *ZINC subset*.

**Dataset Splits** For the citation networks (Cora, CiteSeer, and PubMed), we follow the standard fixed splits introduced in the GCN paper Kipf & Welling (2017), using 20 nodes per class for training (e.g., 140 for Cora, 120 for CiteSeer, and 60 for PubMed), 500 nodes for validation, and 1,000 nodes for testing.

For the PPI dataset, we follow the inductive setting introduced by the GCN paper Kipf & Welling (2017), which uses the Homo sapiens protein–protein interaction (PPI) subgraph with approximately 3,890 nodes and 50 labels, split into 20 graphs for training, 2 graphs for validation, and 2 graphs for testing.

For the ZINC dataset, we use the standard subset split provided by PyG: 12,000 molecules split into 10,000/1,000/1,000 for train/validation/test. We apply random walk positional encodings of length 20 as a pre-transform step, consistent with prior work.

## C   EXPERIMENTAL DETAILS

### C.1   DETAILED RESULTS FOR NODE REPRESENTATION METHODS

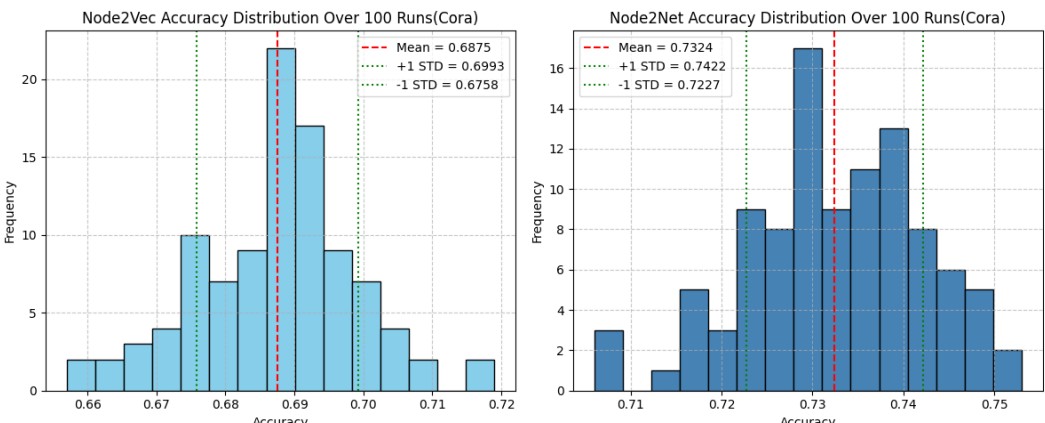

Figure 3: Accuracy distribution over 100 runs for Node2Vec (left) and Node2Net (right) on the Cora dataset. Node2Net shows a tighter and higher performance distribution.

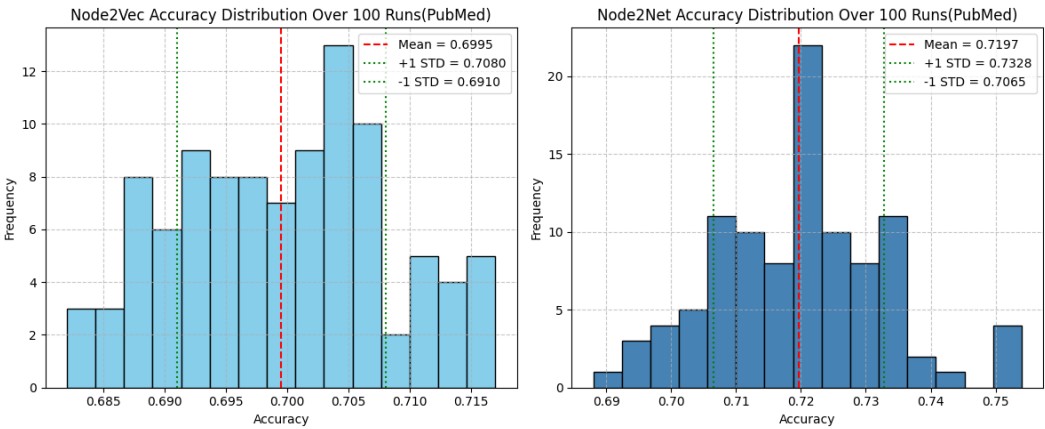

Figure 4: Accuracy distribution over 100 runs for Node2Vec (left) and Node2Net (right) on the PubMed dataset. Node2Net shows a tighter and higher performance distribution.

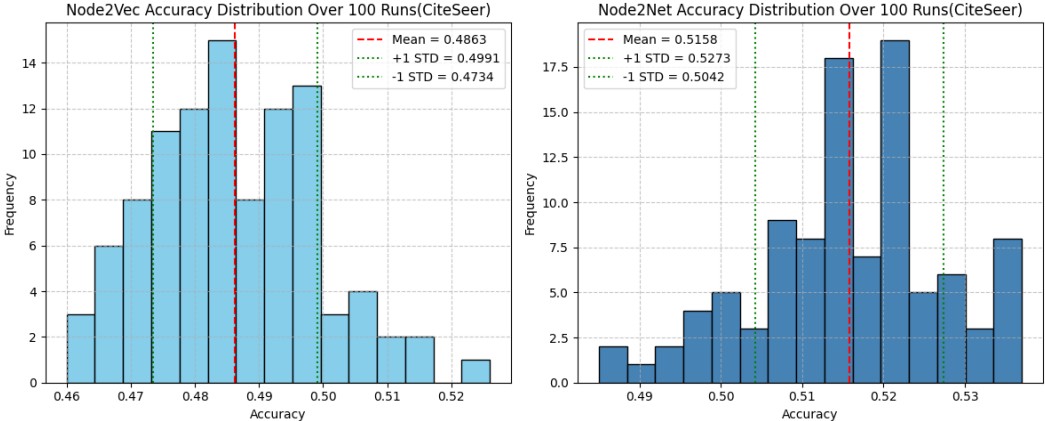

Figure 5: Accuracy distribution over 100 runs for Node2Vec (left) and Node2Net (right) on the CiteSeer dataset. Node2Net shows a tighter and higher performance distribution.

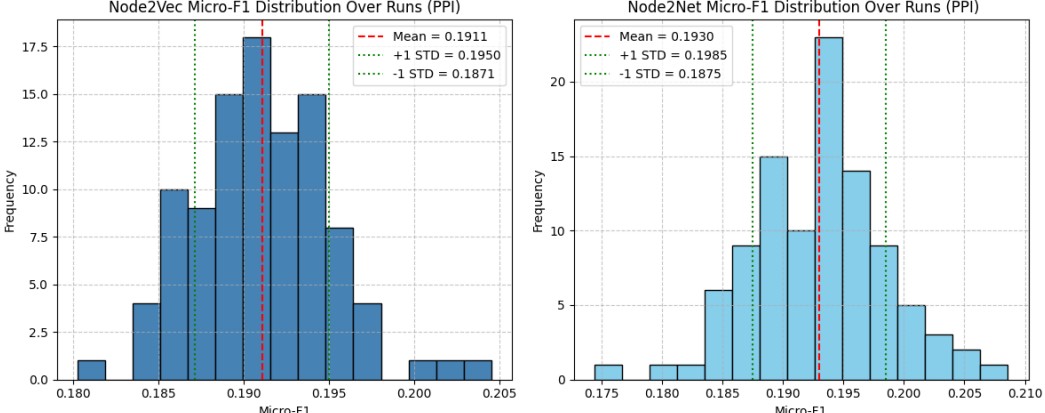

Figure 6: F1 distribution over 100 runs for Node2Vec (left) and Node2Net (right) on the PPI dataset. Node2Net shows a tighter and higher performance distribution.

## C.2 DETAILED RESULTS FOR TRADITIONAL GNNS

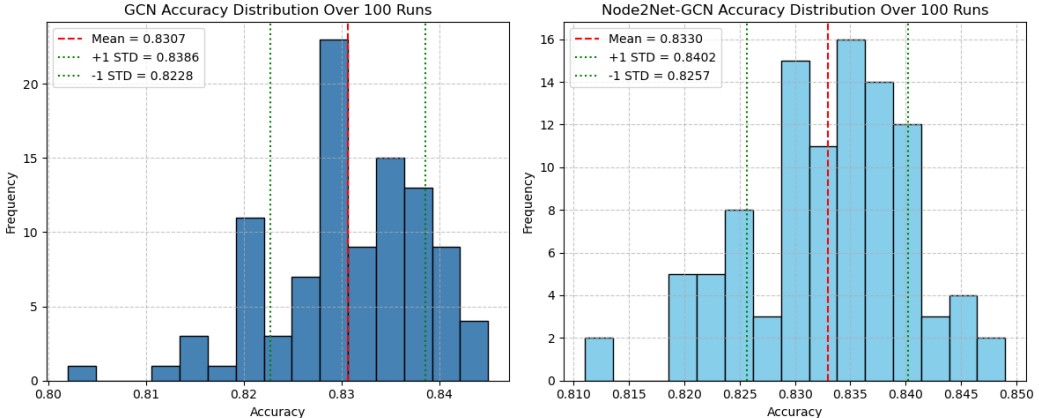

Figure 7: Accuracy distribution over 100 runs on the Cora dataset, using the test results from the last training epoch. GCN (left) and Node2Net-GCN (right) are compared, with Node2Net-GCN showing a tighter and higher performance distribution.

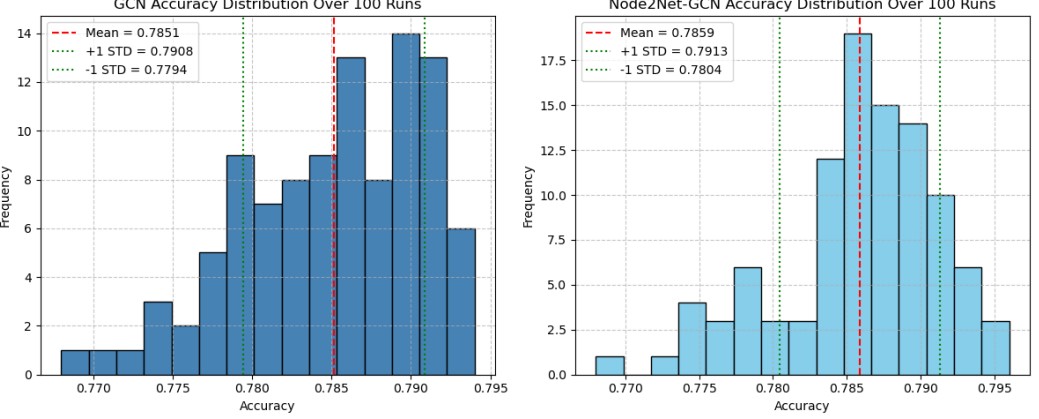

Figure 8: Accuracy distribution over 100 runs on the PubMed dataset, using the test results from the last training epoch. GCN (left) and Node2Net-GCN (right) are compared, with Node2Net-GCN showing a tighter and higher performance distribution.

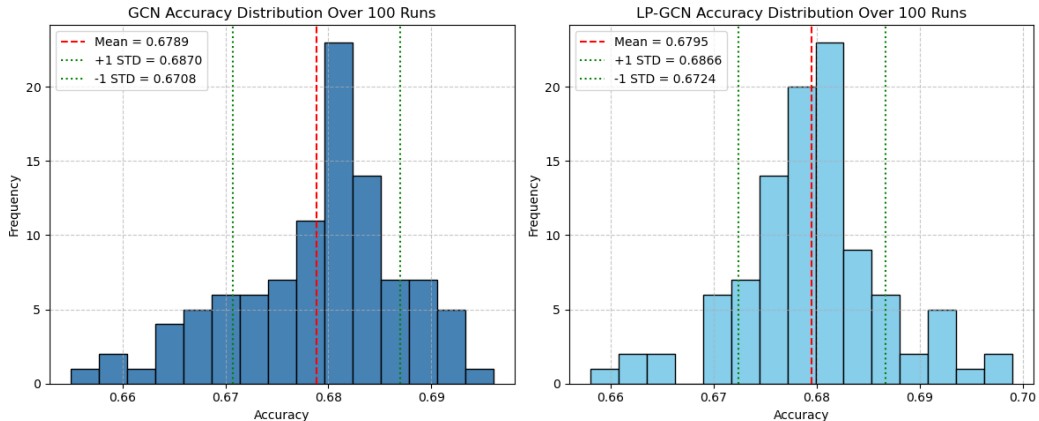

Figure 9: Accuracy distribution over 100 runs on the CiteSeer dataset, using the test results from the last training epoch. GCN (left) and Node2Net-GCN (right) are compared, with Node2Net-GCN showing a tighter and higher performance distribution.

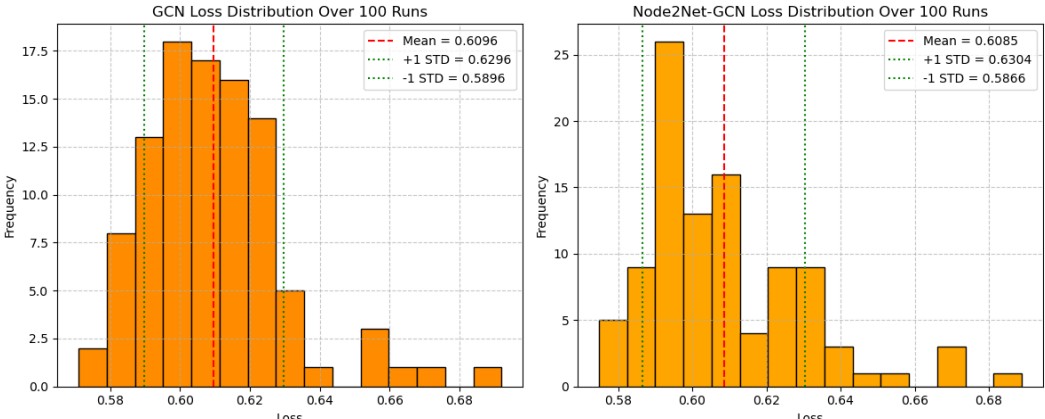

Figure 10: Loss distribution over 100 runs on the Cora dataset, using the test results from the last training epoch. GCN (left) and Node2Net-GCN (right) are compared, with Node2Net-GCN showing a tighter and lower loss distribution.

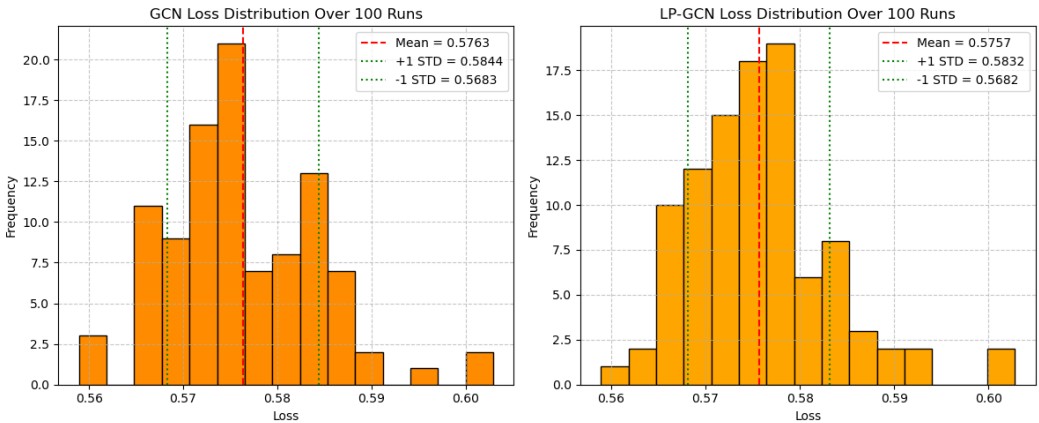

Figure 11: Loss distribution over 100 runs on the PubMed dataset, using the test results from the last training epoch. GCN (left) and Node2Net-GCN (right) are compared, with Node2Net-GCN showing a tighter and lower loss distribution.

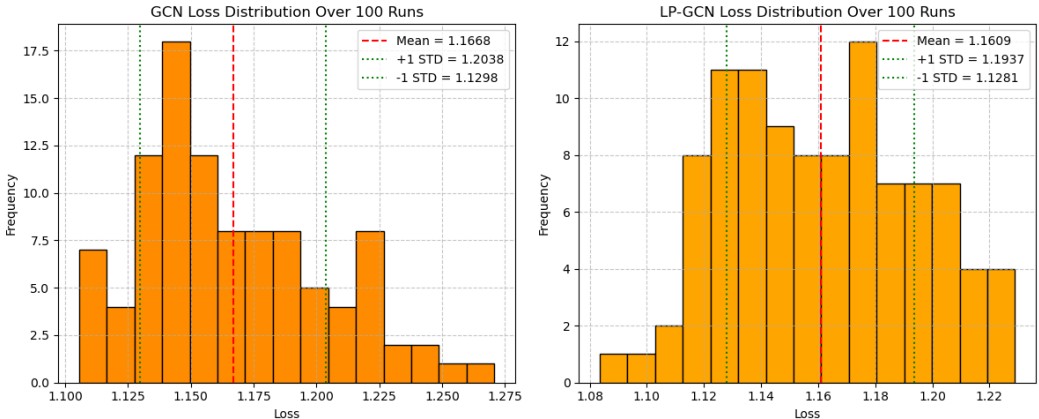

Figure 12: Loss distribution over 100 runs on the CiteSeer dataset, using the test results from the last training epoch. GCN (left) and Node2Net-GCN (right) are compared, with Node2Net-GCN showing a tighter and lower loss distribution.

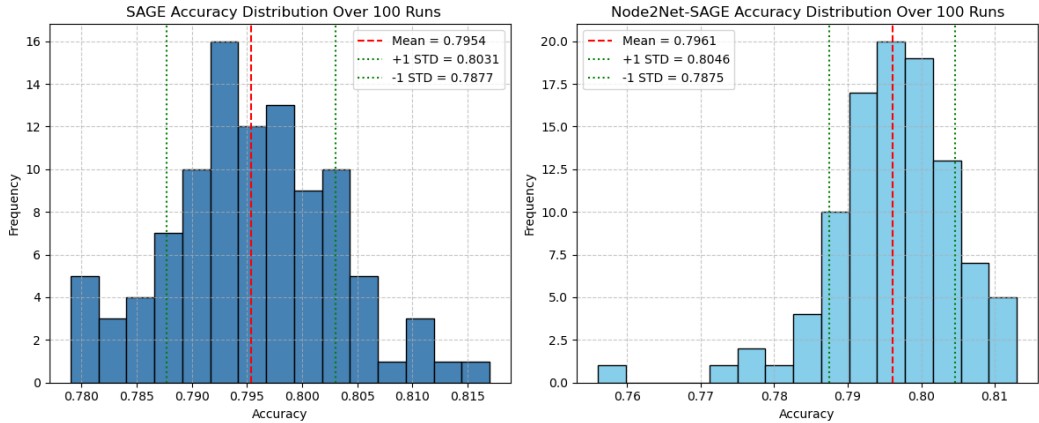

Figure 13: Accuracy distribution over 100 runs on the Cora dataset, using the test results from the last training epoch. GraphSAGE (left) and Node2Net-SAGE (right) are compared, with Node2Net-SAGE showing a tighter and higher performance distribution.

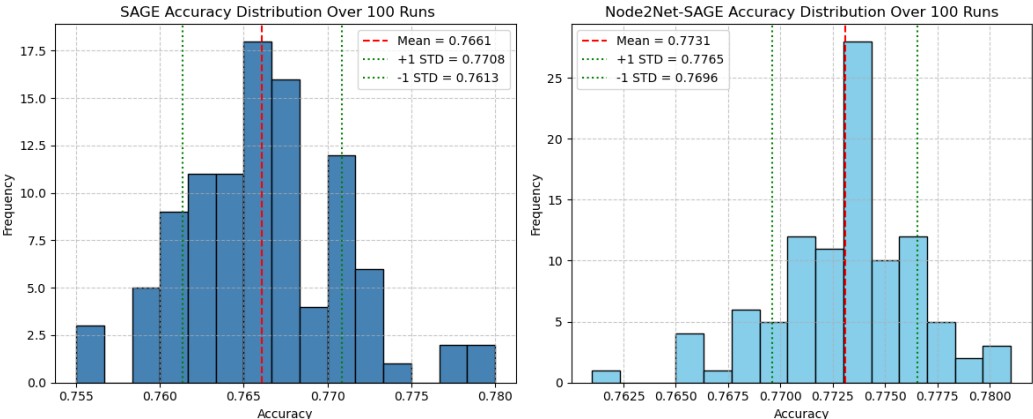

Figure 14: Accuracy distribution over 100 runs on the PubMed dataset, using the test results from the last training epoch. GraphSAGE (left) and Node2Net-SAGE (right) are compared, with Node2Net-SAGE showing a tighter and higher performance distribution.

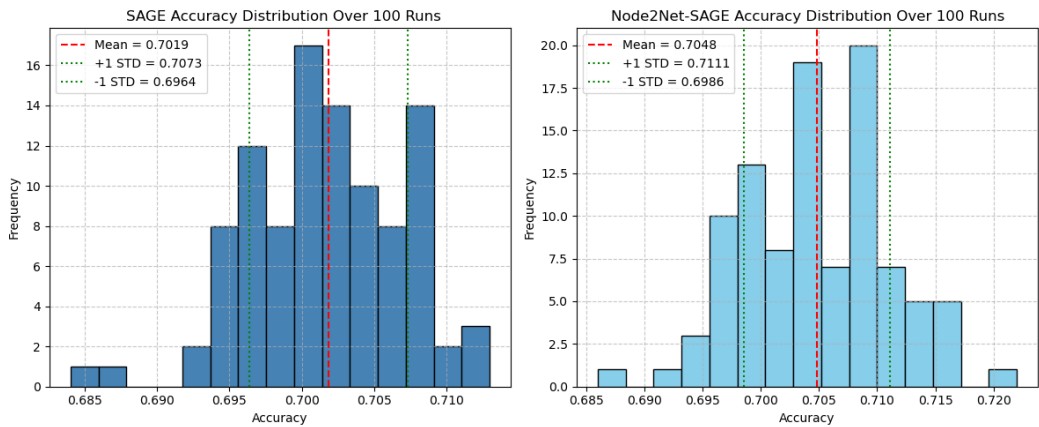

Figure 15: Accuracy distribution over 100 runs on the CiteSeer dataset, using the test results from the last training epoch. GraphSAGE (left) and Node2Net-SAGE (right) are compared, with Node2Net-SAGE showing a tighter and higher performance distribution.

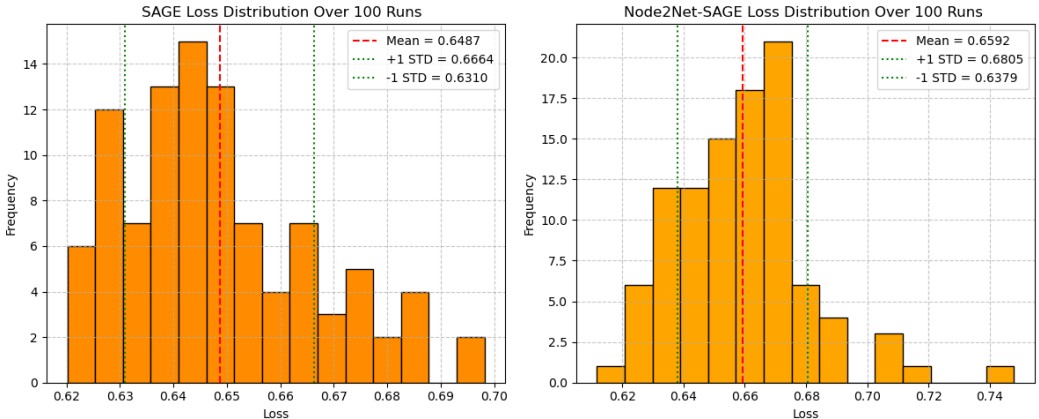

Figure 16: Loss distribution over 100 runs on the Cora dataset, using the test results from the last training epoch. GraphSAGE (left) and Node2Net-SAGE (right) are compared, with Node2Net-SAGE showing a tighter and lower loss distribution.

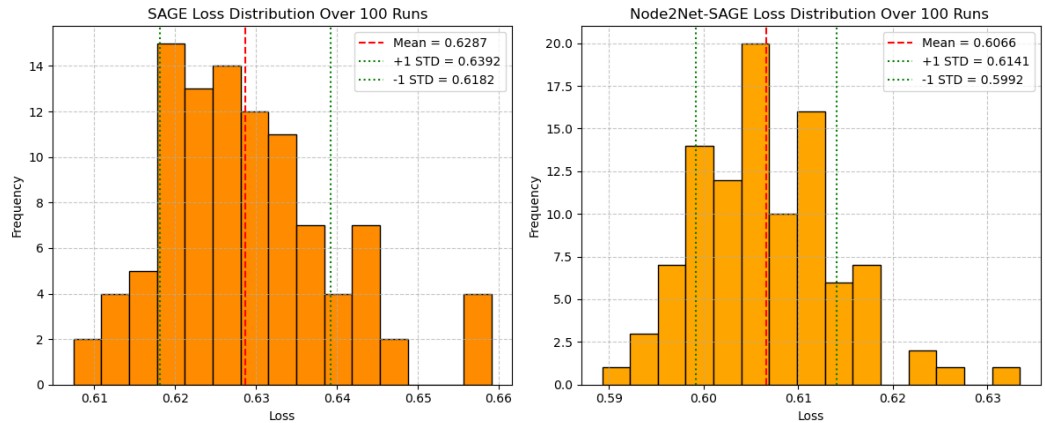

Figure 17: Loss distribution over 100 runs on the PubMed dataset, using the test results from the last training epoch. GraphSAGE (left) and Node2Net-SAGE (right) are compared, with Node2Net-SAGE showing a tighter and lower loss distribution.

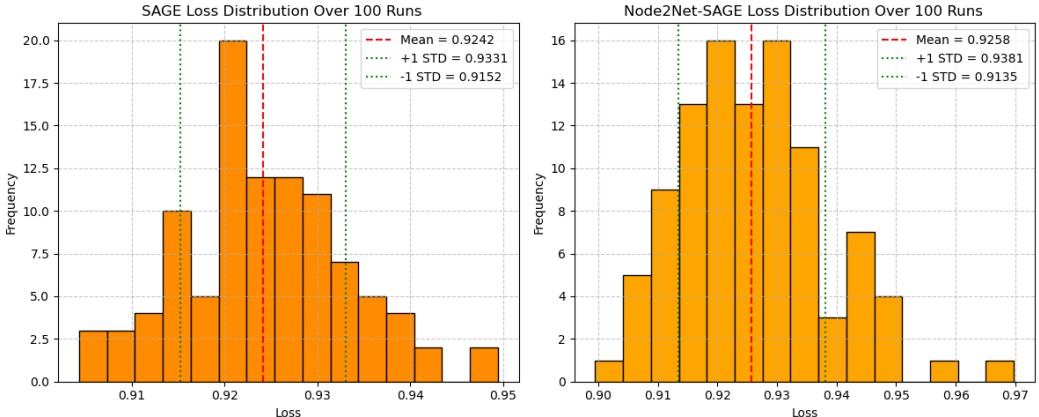

Figure 18: Loss distribution over 100 runs on the CiteSeer dataset, using the test results from the last training epoch. GraphSAGE (left) and Node2Net-SAGE (right) are compared, with Node2Net-SAGE showing a tighter and lower loss distribution.

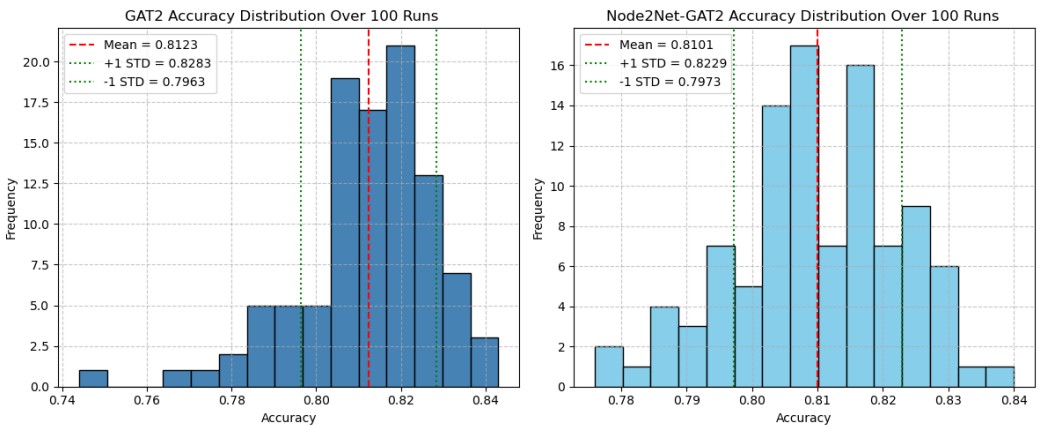

Figure 19: Accuracy distribution over 100 runs on the Cora dataset, using the test results from the last training epoch. GATv2 (left) and Node2Net-GATv2 (right) are compared.

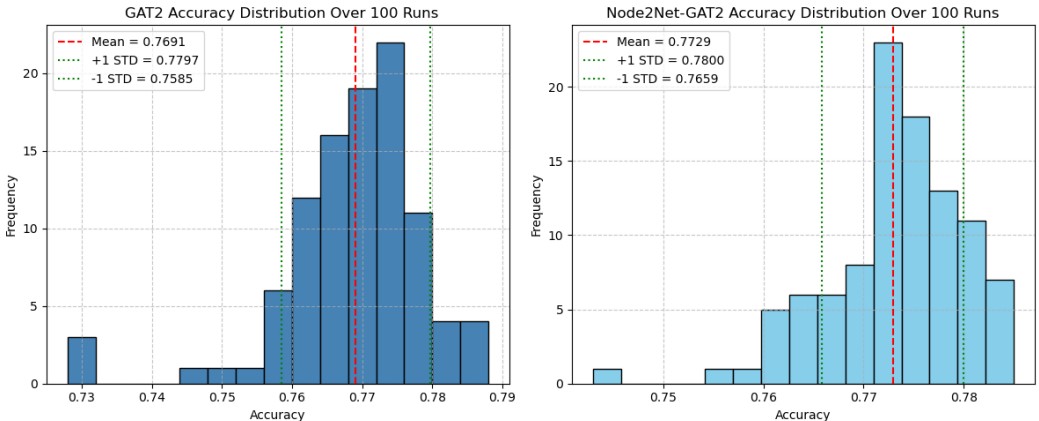

Figure 20: Accuracy distribution over 100 runs on the PubMed dataset, using the test results from the last training epoch. GATv2 (left) and Node2Net-GATv2 (right) are compared, with Node2Net-GATv2 showing a tighter and higher performance distribution.

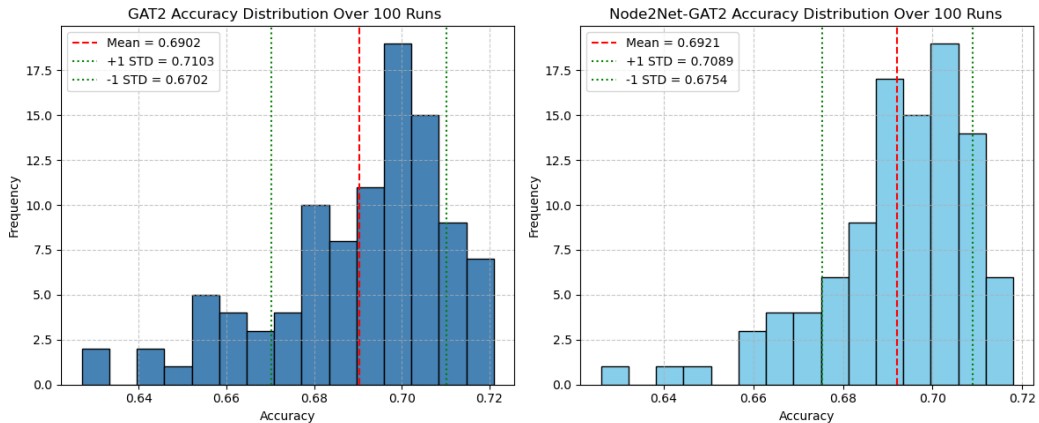

Figure 21: Accuracy distribution over 100 runs on the CiteSeer dataset, using the test results from the last training epoch. GATv2 (left) and Node2Net-GATv2 (right) are compared, with Node2Net-GATv2 showing a tighter and higher performance distribution.

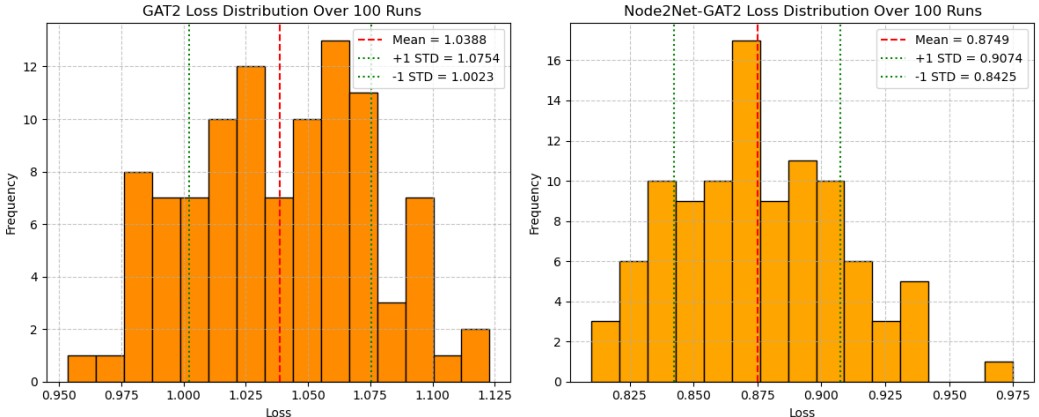

Figure 22: Loss distribution over 100 runs on the Cora dataset, using the test results from the last training epoch. GATv2 (left) and Node2Net-GATv2 (right) are compared, with Node2Net-GATv2 showing a tighter and lower loss distribution.

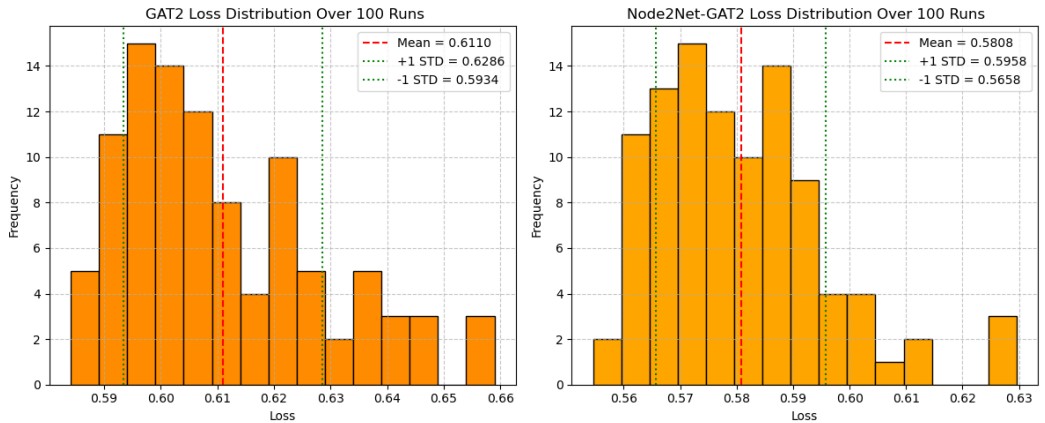

Figure 23: Loss distribution over 100 runs on the PubMed dataset, using the test results from the last training epoch. GATv2 (left) and Node2Net-GATv2 (right) are compared, with Node2Net-GATv2 showing a tighter and lower loss distribution.

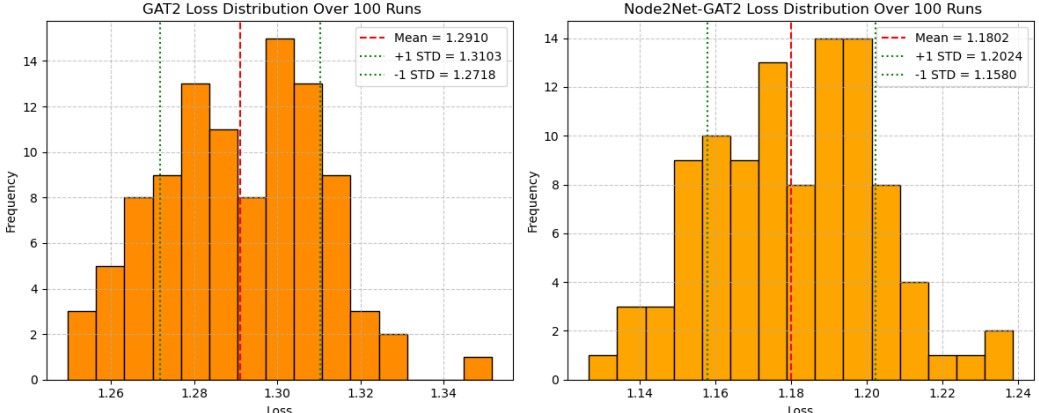

Figure 24: Loss distribution over 100 runs on the CiteSeer dataset, using the test results from the last training epoch. GATv2 (left) and Node2Net-GATv2 (right) are compared, with Node2Net-GATv2 showing a tighter and lower loss distribution.

## C.3 DETAILED RESULTS FOR GRAPH TRANSFORMER METHOD GRAPHGPS

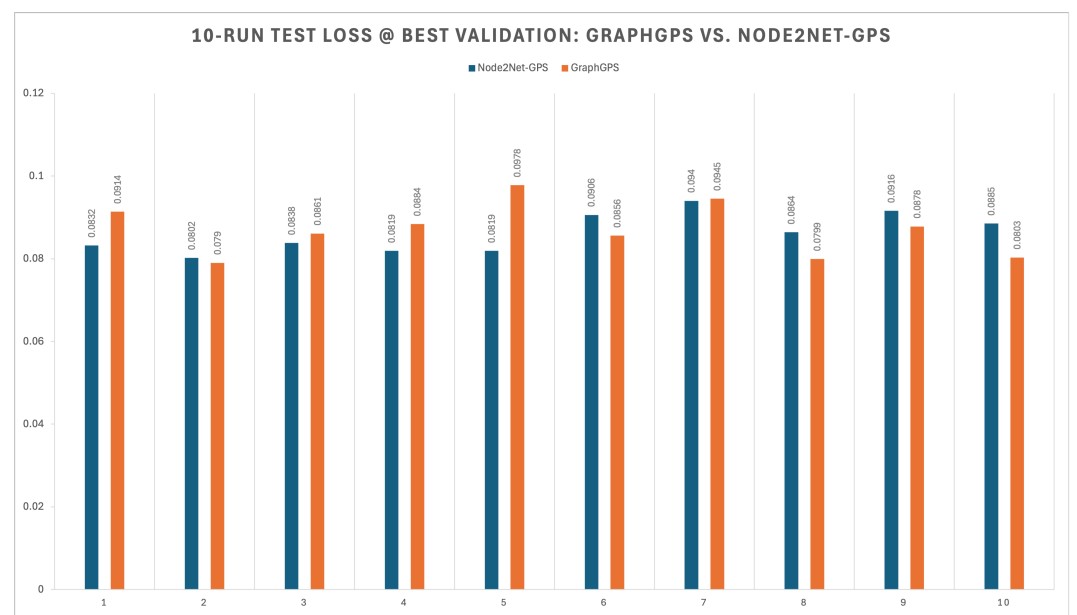

Figure 25: Test loss at best validation across 10 runs for GraphGPS (orange) and Node2Net-GraphGPS (blue) on the ZINC dataset. Each bar represents the loss from one run (2000 training epochs). Node2Net-GraphGPS shows a generally lower and more stable loss compared to the original GPS.

## C.4 COMPUTING ENVIRONMENT

All models are implemented using PyTorch Geometric and trained on an NVIDIA L40S GPU. We evaluate embeddings on the node classification task, where a logistic regression classifier is trained on the learned embeddings. Following standard practice, datasets are split into training/validation/test sets, and accuracy (and F1 score for PPI) is reported. Each experiment is repeated with 100 random seeds, and we report the mean and standard deviation.

All models are implemented using PyTorch 2.3.1.post300, and we use the original GCN implementation provided by the authors at `https://github.com/tkipf/pygcn`. All experiments are conducted on a computer server equipped with one NVIDIA RTX A6000 GPU (48GB memory) and an Intel Xeon w5-2445 CPU (20 cores).

## C.5 HYPERPARAMETERS

Table 5: Hyperparameters for Node2Net-GCN. Citation datasets (Cora, PubMed, and CiteSeer) use Adam optimizer with negative log-likelihood loss (`F.nll_loss`).

| Category | Parameter | Cora | PubMed | CiteSeer | PPI |
|---|---|---|---|---|---|
| Optimization | Node Weights LR | 0.001 | 0.001 | 0.001 | $3\times10^{-5}$ |
| | Node Weights WD | 0.0003 | 0.0003 | 0.0003 | 0.0001 |
| | GCN LR | 0.01 | 0.01 | 0.01 | 0.01 |
| | GCN WD | 0.0001 | 0.0001 | 0.0001 | 0.0005 |
| Architecture | Hidden Units | 16 | 16 | 16 | 128 |
| | Node-MLP Hidden | [32, 32] | [32, 32] | [32, 32] | [64] |
| Training | Dropout | 0.5 | 0.5 | 0.5 | 0.5 |
| | Epochs | 200 | 200 | 200 | 200 |
| | Runs | 100 | 100 | 100 | 100 |

Table 6: Hyperparameters for Node2Net-GraphSAGE

| Category | Parameter | PubMed | Cora | CiteSeer | PPI |
|---|---|---|---|---|---|
| Optimization | V-MLP LR | $1\times10^{-5}$ | $1\times10^{-5}$ | $2\times10^{-5}$ | $1\times10^{-5}$ |
| | V-MLP Weight Decay | 0.0003 | 0.0 | 0.0005 | 0.0005 |
| | SAGE LR | 0.01 | 0.01 | 0.01 | 0.01 |
| | SAGE Weight Decay | 0.0005 | 0.0005 | 0.0005 | 0.0005 |
| Architecture | Hidden Units | 16 | 16 | 16 | 128 |
| | V-MLP Hidden | [8, 8] | [32] | [32, 32] | [64] |
| | Hops | 2 | 4 | 1 | – |
| Training | Dropout | 0.15 | 0.0 | 0.0 | 0.5 |
| | Epochs | 200 | 200 | 400 | 200 |
| | Runs | 100 | 100 | 100 | 100 |

Table 7: Hyperparameters for Node2Net-GATv2. All citation datasets (Cora, PubMed, and Cite-Seer) are preprocessed using `NormalizeFeatures` from `torch_geometric.transforms`.

| Category | Parameter | PubMed | Cora | CiteSeer | PPI |
|---|---|---|---|---|---|
| Optimization | Node-MLP LR | $1\times10^{-6}$ | $1\times10^{-6}$ | $1\times10^{-6}$ | $5\times10^{-6}$ |
| | Node-MLP WD | 0.0 | 0.0 | 0.0 | 0.0 |
| | GATv2 LR | 0.01 | 0.01 | 0.01 | 0.01 |
| | GATv2 WD | 0.0005 | 0.0005 | 0.0005 | 0.0005 |
| Architecture | Heads | 1 | 1 | 1 | 1 |
| | Hidden Units | 16 | 16 | 16 | 16 |
| | Hops | 1 | 1 | 1 | – |
| Dropout | | 0.5 | 0.5 | 0.5 | 0.6 |
| Training | Epochs | 200 | 200 | 200 | 200 |
| | Runs | 100 | 100 | 100 | 100 |

Table 8: Hyperparameters for Node2Net-GPS

| Category | Parameter | Value |
|---|---|---|
| Training | Runs | 10 |
| | Epochs | 2000 |
| | Edge node MLP update ratio | 5 |
| Optimization | Learning Rate (LR) | 0.001 |
| | Weight Decay | $1\times10^{-5}$ |
| | LR Patience | 20 |
| | Min LR | $1\times10^{-5}$ |
| | LR Factor | 0.5 |
| | Dropout | 0.0 |
| Architecture | Channels | 64 |
| | Positional Enc. Dim. | 8 |
| | Num. Layers | 10 |
| | Attention Type | multihead |
| | Attention Heads | 4 |
| | Attention Dropout | 0.5 |

**Random Seeds and Reproducibility.** To ensure fair and reproducible comparisons, we adopted dataset- and model-specific random seed settings following prior work. For the citation benchmarks (Cora, CiteSeer, PubMed) and the PPI dataset, we trained each of the **GCN**, **GraphSAGE**, and **GATv2** models over **100 independent runs**, with random seeds uniformly sampled from **1 to 100**. For the **GPS** model on the **ZINC** dataset, we followed the experimental protocol from the original

*"Recipe for a General, Powerful, Scalable Graph Transformer"* paper, performing **10 runs** of **2000 training epochs** each, with random seeds ranging from **1 to 10**. For **HoloGNN**, we reproduced the setting described in its original paper **?**, which reports results over three random seeds; accordingly, we conducted **3 runs** using seeds **42, 88, and 456**.

# D    NODE2NET MLP VARIANTS

To explore the impact of node-specific feature transformations, we design several Node2Net Multi-Layer Perceptron (MLP) variants tailored for different learning tasks. These modules replace or augment standard embedding layers, serving as flexible pre-transformations of node and edge features. We summarize three representative designs below.

**(1) Vanilla Node2Net MLP (V-MLP):**    The basic variant is a feed-forward MLP applied directly to input node attributes $X_i \in \mathbb{R}^F$. It consists of an input projection, one or more hidden layers with ReLU activation, and an output projection:

$$h_i = \text{V-MLP}(X_i).$$

This architecture provides a straightforward non-linear mapping, and we employ it in node classification experiments as a lightweight feature extractor.

**(2) Residual Node2Net MLP (R-MLP):**    To enhance gradient flow and mitigate vanishing effects, we implement a residual version where the input is added back to the MLP output:

$$h_i = X_i + \text{R-MLP}(X_i).$$

This skip connection allows the model to preserve raw features while learning refinements, improving stability during node classification tasks.

**(3) NodeEdge2Net MLP (NE-MLP):**    For graph-level prediction tasks, we extend the idea of per-node MLPs to encompass both node types and edge types. NE-MLP replaces traditional embedding layers with categorical MLPs operating on one-hot identifiers, jointly with a projection for positional encodings (PE). Formally,

$$h_i = \left[ \, \phi_{\text{node}}(\text{onehot}(t_i)) \,\|\, \phi_{\text{PE}}(\text{PE}_i) \, \right],$$
$$e_{ij} = \phi_{\text{edge}}(\text{onehot}(r_{ij})),$$

where $t_i$ is the node type, $r_{ij}$ is the edge type, and $\phi_{\text{node}}, \phi_{\text{edge}}, \phi_{\text{PE}}$ are MLPs. Gradient scheduling is further introduced to decouple node and edge updates: node MLPs update every iteration, while edge MLPs update at a configurable frequency. This design is particularly suited for graph-level tasks (e.g., molecular property prediction), where structured categorical information and positional encodings must be fused into channel-aligned features.

# E    THE USE OF LARGE LANGUAGE MODELS

We used Large Language Models to correct typos and syntax errors.

