# OpenReview forum: "Node2Net: Node-Specific Parameterization for Expressive Graph Representation Learning"
_ICLR.cc/2026/Conference — Submitted to ICLR 2026_

### Official Review · Reviewer_gMZx · 2025-10-18

**Soundness:** 2
**Presentation:** 2
**Contribution:** 2
**Rating:** 2
**Confidence:** 5

**Summary:**

This paper introduces Node2Net, a GNN framework where each node is represented by its own small neural network instead of a shared embedding. This design enhances expressiveness beyond the 1-WL limit while remaining computationally efficient. Node2Net is model-agnostic and consistently outperforms traditional GNNs and Graph Transformers across benchmarks.

**Strengths:**

1. The intuition behind Node2Net is simple, intuitive, and easy to follow. It is model-agnostic and can be seamlessly adapted to different graph learning architectures and tasks.
2. The paper provides numerous examples demonstrating the applicability of Node2Net, including its integration with MPNNs and Graph Transformers, highlighting the elegance and versatility of the approach.
3. The experimental section is comprehensive, with detailed settings and clear explanations of results, effectively showing the advantages and practical impact of Node2Net.

**Weaknesses:**

1. Although the intuition behind Node2Net is simple, it feels somewhat trivial and insufficiently developed. I think the paper should carefully analyze the trade-off between performance gains and the substantial costs in terms of parameters and computational complexity. Node2Net introduces significant overhead, making it difficult to scale to large graphs—this issue must be explicitly discussed.
2. The performance improvement brought by Node2Net appears limited. As shown in Table 2, the gains over backbone models are minor, and the paper does not compare against recent state-of-the-art baselines.
3. The paper does not provide publicly available code for reproduction, which is an essential requirement for modern learning-based research.
4. The writing quality requires further improvement; see the minor comments below. Additionally, Table 3 reports loss values directly, which is unconventional and potentially confusing; it would be clearer to present convergence curves over time or epochs instead.

**Minor Comments:**
(1) The figures in the paper are not vector graphics, which do not conform to academic publication standards.
(2) Figure 2 is difficult to understand, and it is unclear which module the term “Node2Net” represents.
(3) The mathematical notation in the paper is not clear. Vectors and matrices should be typeset in bold for clarity. It is recommended to follow the formatting guidelines in the official ICLR template.

**Questions:**

1. Please begin by responding to the Weaknesses part.
2. Please discuss the overhead introduced by Node2Net in terms of parameters and computational complexity, especially from an empirical perspective. Is it feasible to scale to very large graphs (e.g., with hundreds of millions of nodes)? It is worth noting that Node2Vec can handle such scales.

---

> ### Author Response · Authors · 2025-11-14
> **response to the 4 weaknesses and 2 questions**
>
> Weakness 1: The contributions of our Node2Net approach are two-fold: theoretically we propose a more reasonable way to id a node and breaks 1-WL indistinguishability than current NodeID methods; at the model architecture level, Node2Net expands the design space of GNN proposed in https://proceedings.neurips.cc/paper/2020/file/c5c3d4fe6b2cc463c7d7ecba17cc9de7-Paper.pdf. Performance-wsie, the improvement is small but consistent across multiple benchmarks whose performance has been stalled for a long time. The small improvement is due to the data split, for example, in Cora only 140 out of 2708 nodes (5%) are trained with Node2Net. In node-level tasks, due to the constraint of data split (nodes in testing set can not be trained in training set), node representation methods including Node2Net will better fit in subgraph-level and graph-level tasks, which we will explore next. In this work, we still want to show Node2Net's effectiveness on the fundamental node level, even if it is not the optimal setting for node representation methods. We will expand to large-scale graphs in the next step.
>
> Weakness 2: The performance in the GNN field has been stalled in the last 7-8 years actually since GCN in 2017. Take Cora dataset as an example, according to the Cora leaderboard https://hyper.ai/en/sota/tasks/node-classification/benchmark/node-classification-on-cora, it seems that recent methods made a significant improvement on performance (around 90% now over GCN's ~81%). However these recent methods often adopted a different split (70%-80% of 2708 nodes in training set vs. 140 out of 2708 nodes (5%) in training in original GCN setting). Using the 5% original split in GCN paper, probably none of GNN papers in the last 7-8 years can beat the GCN performance (orignal GCN paper reported ~81% Cora accuracy, but based on our experiments, GCN's performance is around ~83%). All of our experiments are run 100 times to make sure our improvement is real and not due to randomness in training.
>
> The most fair baseline to compare with is existing node representation methods, and Node2Net shows significant improvement over the classic Node2Vec.
>
> We chose the most influential GNN methods as baselines since most recent state-of-the-art methods are based on them. By choosing these broadly-applied methods, we are confident that Node2Net can be applied and bring improvement with recent methods that build upon these classical methods.
>
> Weakness 3: Code will be made available upon publication.
>
> Weakness 4: Thanks for the detailed comments, we will fix them in the revised version. Loss is an important measure for machine learning methods since it usually directly optimized. Our experiments on losses do show significant improvement over baselines.
>
> Question 1: thanks.
>
> Question 2: Node2Net will use more memory to hold MLP parameters for each node. Take Cora dataset as an example, it has 2708 nodes, each node has 1433 features. If we insert a 1-hidden-layer of size 8 into a node, for each node, the number of added parameters will be 1433 x 8 + 8 x 7 = 11520. For 2708 nodes, the total number of added parameters is ~31M. However, one alternative is to apply Node2Net only to nodes appearing in training set (140 out of 2708 in Cora dataset), in this case the total number of added parameters is ~1.6M. Another option is to lower the number of input features, which can bring significant savings. Take Cora as an example, in the 1433 input features of Cora, usually there are around 2-50 non-zero values (showing occurrences of unique words), which means that the input data is probably very low rank. If we embed one original 1433-feature vector to an embedding of 50, the total number of added parameters is 56K. Take a large graph with 1M nodes in training set, Node2Net will need 400M extra parameters, which still can be comfortably processed by even a consumer-grade GPU. Usually the running time roughly doubles comparing with base model. We will include all these details in the revised version.

---

> > ### Comment · Reviewer_gMZx · 2025-11-18
> >
> > Thank you for the response. I have carefully reviewed the rebuttal; however, it does not adequately address my concerns. The simple example on the Cora dataset is not convincing. To demonstrate the scalability of Node2Net, the authors should provide theoretical analyses of its time and space complexity on large-scale graphs, as well as experimental results on graphs containing tens of millions of nodes. Therefore, I will maintain my original score.

---

### Official Review · Reviewer_WvU1 · 2025-10-24

**Soundness:** 2
**Presentation:** 3
**Contribution:** 1
**Rating:** 2
**Confidence:** 4

**Summary:**

The paper proposes including node specific weight parameters in the GNN model. Experimental results evaluate the proposed approach with various GNN backbones.

**Strengths:**

* The paper is well written.
* Broad Applicability: The method is model-agnostic and can be integrated into various GNN architectures (node representation methods, MPNNs, Graph Transformers), demonstrating versatility.
* Theoretical Contributions: The paper provides theoretical analysis showing that Node2Net can break 1-WL indistinguishability and is strictly more expressive than static node embeddings.

**Weaknesses:**

* While consistent, the improvements shown in Tables 1-4 are often marginal (e.g., 83.07% → 83.30% for GCN on Cora). This raises questions about the practical significance of the added complexity.
* While the paper claims linear scaling, the actual runtime and memory overhead compared to baselines is not reported. Each node now requires storing and updating an entire MLP.
* The experiments focus on relatively small graphs (largest is PubMed with ~20K nodes). Scalability to million-node graphs is unclear.
* While the paper claims Node2Net doesn't affect inductive properties, the pre-training step for unseen nodes seems problematic. How well does this work for completely new nodes at test time?
* The paper dismisses node-ID methods as having "randomness and instability" but doesn't provide empirical comparisons to support this claim.
* The paper doesn't provide insights into what types of graphs or tasks benefit most from node-specific parameterization.

**Questions:**

* Can you provide concrete runtime and memory comparisons? For a graph with 1M nodes, what is the actual memory footprint of storing 1M MLPs?
* Given that Node2Net can break 1-WL, why are the empirical improvements so modest? Are the benchmarks not challenging enough to showcase the theoretical advantages?
* How exactly does the pre-training step work for nodes that appear only at test time? What if these nodes have feature distributions very different from training nodes?
* Can you include empirical comparisons with random node features/IDs to substantiate your claims about their instability?
* Are there specific graph properties (e.g., heterophily, specific structural patterns) where Node2Net shows more significant improvements?
*

---

> ### Author Response · Authors · 2025-11-14
> **response to the 6 weaknesses and 5 questions**
>
> Weakness 1: The performance in the GNN field has been stalled in the last 7-8 years actually since GCN in 2017. Take Cora dataset as an example, according to the Cora leaderboard https://hyper.ai/en/sota/tasks/node-classification/benchmark/node-classification-on-cora, it seems that recent methods made a significant improvement (around 90% now over GCN's ~81%). However these recent methods often adopted a different split (70%-80% of 2708 nodes in training set vs. 140 out of 2708 nodes (5%) in training in original GCN setting). Using the 5% original split in GCN paper, probably none of GNN papers in the last 7-8 years can beat the GCN performance (original GCN paper reported ~81% Cora accuracy, but based on our experiments, GCN's performance is around ~83%). All of our experiments are run 100 times to make sure our improvement is real and not due to randomness in training. Loss is an important measure for machine learning methods since it usually directly optimized. Our experiments on losses do show significant improvement over baselines.
>
> The most fair baseline to compare with is existing node representation methods, and Node2Net shows significant improvement over the classic Node2Vec.
>
> The contributions of our Node2Net approach are two-fold: theoretically we propose a more reasonable way to id a node and breaks 1-WL indistinguishability than current NodeID methods; at the model architecture level, Node2Net expands the design space of GNN proposed in https://proceedings.neurips.cc/paper/2020/file/c5c3d4fe6b2cc463c7d7ecba17cc9de7-Paper.pdf. Performance-wsie, the improvement is small but consistent across multiple benchmarks whose performance has been stalled for a long time. The small improvement is due to the data split, for example, in Cora only 140 out of 2708 nodes (5%) are trained with Node2Net.
>
> Weakness 2: Node2Net will use more memory to hold MLP parameters for each node. Take Cora dataset as an example, it has 2708 nodes, each node has 1433 features. If we insert a 1-hidden-layer of size 8 into a node, for each node, the number of added parameters will be 1433 x 8 + 8 x 7 = 11520. For 2708 nodes, the total number of added parameters is ~31M. However, one alternative is to apply Node2Net only to nodes appearing in training set (140 out of 2708 in Cora dataset), in this case the total number of added parameters is ~1.6M. Another option is to lower the number of input features, which can bring significant savings. Take Cora as an example, in the 1433 input features of Cora, usually there are around 2-50 non-zero values (showing occurrences of unique words), which means that the input data is probably very low rank. If we embed one original 1433-feature vector to an embedding of 50, the total number of added parameters is 56K. Take a large graph with 1M nodes in training set, Node2Net will need 400M extra parameters, which still can be comfortably processed by even a consumer-grade GPU. Usually the running time roughly doubles comparing with base model. We will include all these details in the revised version.
>
> Weakness 3: As discussed above, two options exist to reduce the number of extra parameters: only apply Node2Net to nodes in the training set, and adopt a low-rank embedding. Due to the computation constraint, we were not able to try with more datasets. We will include more experiment results in the revised version.
>
> Weakness 4: In node-level tasks, due to the constraint of data split, new nodes in testing set can not be trained, which is the general limitation for the whole node representation field. Node representation methods including Node2Net will better fit in subgraph-level and graph-level tasks, which we will explore next. In this work, we still want to show Node2Net's effectiveness on the fundamental node level, even if it is not the optimal setting for node representation methods.
>
> Weakness 5: The instability is widely recognized in the field. For example, please refer to https://arxiv.org/pdf/2005.10039.
>
> Weakness 6: Node2Net can capture fine-grained node-specific patterns and distinguish graphs that are 1-WL-indistinguishable. In this sense, Node2Net will be most helpful for isomorphic graphs.
>
> Question 1: same response to Weakness 2.
>
> Question 2: same response to Weakness 1.
>
> Question 3: Due to a dataset splitting to training, validation, and testing sets, some nodes appearing in testing set do not appear in training set. Node2Net pretrains all nodes with identity mapping makes sure nodes only appearing in test set will input and also output the original features since they will not be trained during training phase of Node2Net. Shifting of node distribution will be a challenge for node representation methods, which have to rely on higher-level learning for mitigation.
>
> Question 4: Thanks for the advice. Due to the computation constraint, we were not able to try with more experiments. We will include more experiment results in the revised version.
>
> Question 5: same as Weakness 6.

---

### Official Review · Reviewer_abjw · 2025-10-31

**Soundness:** 3
**Presentation:** 3
**Contribution:** 1
**Rating:** 2
**Confidence:** 4

**Summary:**

In the submitted manuscript, the authors propose to include a node-wise lightweight MLP in different deep learning methods on graphs, including unsupervised node embedding methods such as node2vec, standard Graph Neural Networks, such as GCNs, GAT and GraphSage and graph transformers such as GraphGPS. The authors show that this addition allows MPNNs to distinguish graphs that are not 1-WL distinguishable. They furthermore provide empirical results on 5 datasets for their different model variants.

**Strengths:**

- The proposed idea has not been explored in the literature on Graph Neural Networks as far as I know.

- The ability to map automorphic nodes to different representations could potentially be interesting if properly explored and analysed.

**Weaknesses:**

- It seems to me that the presented idea does not generalise to unseen nodes and that the authors therefore have to rely on the exclusion of their proposed MLPs for the test set in the case of GNNs and rely on pretraining or gradient scheduling tricks to not harm performance in practice.

- The observed performance improvements are insignificant in the majority of cases: In Table 2 there is only one dataset for which one GNN shows a significant improvement over the baseline. In Table 4 no significant improvement is observed. So, the only improvements you observed were in the context of the node2vec method, where you introduced your lightweight networks only after training a node2vec model in isolation. I am therefore, not convinced of the empirical benefit of your method.

- Certain presentation styles like the lists in Lines 126-151, 301-310 and (especially) 447-455 are reminiscent of the writing style of an LLM, the use of which is not disclosed by the authors. I think it may be better to opt for a more compact form of presentation, which may avoid such suspicion.

**Questions:**

1] I have a fundamental question about your approach: Since MLPs are universal approximators, a global shared MLP (as is standard in GNNs) can learn any well-defined function, which should include almost all functions that your formulation fitting one MLP per node can learn. The only conceptual difference that I see between your approach and the global shared MLP is that you are able to learn functions that are "not well-defined", i.e., functions that map identical inputs to different outputs. This minor difference would also explain why in practice your approach does not yield significant performance improvements on GNNs. Is this right? Or are there further function classes that a global shared MLP cannot learn, which your Node2Net formulation is able to fit?

2] The added time and memory cost of your method in practice is not clear to me. Could you provide these statistics for the results you show in Tables 1, 2 and 4. I suppose since your Node2Net is usually added after a base model is pre-trained, your method should come at an additional training time cost and since it introduces more parameters, it should almost surely increase the memory cost?

3] In Table 2 I am curious what the parameter count of the compared models is? Does your Node2Net variant have more parameters than the base model? And if yes, how does the performance of the base model change when you allocate equally many parameters to the globally shared update step as you do in your Node2Net model?

4] The performance of your graph transformer variant is only measured on one dataset. This seems like an insufficient empirical evidence basis to make any conclusive statements about the improvements of your method. Would it be possible for you to include further datasets in your study of graph transformers?

5] Minor Comments:

5.1] Typo: "nerual" in Line 197

5.2] In Line 243 you define the attention mechanism of graph transformers to only aggregate over neighbourhoods of nodes. To the best of my knowledge, most graph transformers aggregate over the whole node set instead. I think it may be appropriate to edit your formula accordingly, unless you discuss a particular kind of graph transformer here?

5.3] The proof of Theorem 1 has two \qed symbols.

5.4] I found Section 3.5 to have a very low information density. It seems to me that the content of this section could be expressed a lot more concisely. In particular, several of the points mentioned in Lines 301-10 seemed somewhat obvious to me after reading the remainder of your paper.

---

> ### Author Response · Authors · 2025-11-14
> **response to the 3 weaknesses and 5 questions**
>
> Weakness 1: In node-level tasks, due to the constraint of data split (nodes in testing set can not be trained in training set), node representation methods including Node2Net will be a better fit for subgraph-level and graph-level tasks, which we will explore next. In this work, we still want to show Node2Net's effectiveness on the fundamental node level, even if it is not the optimal setting for node representation methods.
>
> Weakness 2: The performance in the GNN field has been stalled in the last 7-8 years actually since GCN in 2017. Take Cora dataset as an example, according to the Cora leaderboard https://hyper.ai/en/sota/tasks/node-classification/benchmark/node-classification-on-cora, it seems that recent methods made a significant improvement on performance (around 90% now over GCN's ~81%). However these recent methods often adopted a different split (70%-80% of 2708 nodes in training set vs. 140 out of 2708 nodes (5%) in training in original GCN setting). Using the 5% original split in GCN paper, probably none of GNN papers in the last 7-8 years can beat the GCN performance (original GCN paper reported ~81% Cora accuracy, but based on our experiments, GCN's performance is around ~83%). The contributions of our Node2Net approach are two-fold: theoretically we propose a more reasonable way to id a node and breaks 1-WL indistinguishability than current node-ID methods; at the model architecture level, Node2Net expands the design space of GNN proposed in https://proceedings.neurips.cc/paper/2020/file/c5c3d4fe6b2cc463c7d7ecba17cc9de7-Paper.pdf. Performance-wise, the improvement is small but consistent across multiple benchmarks whose performance has been stalled for a long time. The small improvement is due to the data split, for example, in Cora only 140 out of 2708 nodes (5%) are trained with Node2Net. All of our experiments are run 100 times to make sure our improvement is real and not due to randomness in training.
>
> The most fair baseline to compare with is existing node representation methods, and Node2Net shows significant improvement over the classic Node2Vec.
>
> Loss is an important measure for machine learning methods since it usually directly optimized. Our experiments on losses do show significant improvement over baselines.
>
> Weakness 3: As disclosed, we used LLM to fix grammar and writing. We will reformat these sections in the revised version.
>
> Question 1: The key idea of Node2Net is to use a node-specific MLP to learn node-specific function/transformation, which is the goal of node-id methods. A shared MLP will not break 1-WL indistinguishability and improve expressiveness.
>
> Question 2: Fair question. Node2Net will use more memory to hold MLP parameters for each node. Take Cora dataset as an example, it has 2708 nodes, each node has 1433 features. If we insert a 1-hidden-layer of size 8 into a node, for each node, the number of added parameters will be 1433 x 8 + 8 x 7 = 11520. For 2708 nodes, the total number of added parameters is ~31M. However, one alternative is to apply Node2Net only to nodes appearing in training set (140 out of 2708 in Cora dataset), in this case the total number of added parameters is ~1.6M. Another option is to lower the number of input features, which can bring significant savings. Take Cora as an example, in the 1433 input features of Cora, usually there are around 20-50 non-zero values (showing occurrences of unique words), which means that the input data is probably very low rank. If we embed one original 1433-feature vector to an embedding of 50, the total number of added parameters is 56K. Take a large graph with 1M nodes in training set, Node2Net will need 400M extra parameters, which still can be comfortably processed by even a consumer-grade GPU. Usually the running time roughly doubles comparing with base model.
>
> Question 3: Node2Net inserts an MLP into each node. So it will use more parameters than a base model. The actually contributions of our Node2Net approach are two-fold: theoretically we propose a more reasonable way to id a node and breaks 1-WL indistinguishability than current NodeID methods; at the model architecture level, Node2Net presents a new direction in GNN architecture design and expands the design space of GNN proposed in https://proceedings.neurips.cc/paper/2020/file/c5c3d4fe6b2cc463c7d7ecba17cc9de7-Paper.pdf.
>
> Question 4: Thanks for the advice. Due to the computation constraint, we were not able to try with more datasets. We will include more experiment results in the revised version.
>
> Question 5: Thanks for the advice. We will fix them in the revised version.

---

> > ### Comment · Reviewer_abjw · 2025-11-20
> >
> > Thank you very much for your fast response. After working through your reply, I have to conclude that your manuscript is not ready for publication yet and I choose to maintain my score therefore. To provide more detail on this decision: W1] I agree, it is possible that your method may exhibit greater advantages in sub-graph or graph-level tasks. I agree that node-level tasks as submitted here are definitely not optimal. W2 & Q2] I don't see a solid argument for roughly doubling the runtime of the model when in most displayed settings in your paper the performance is not significantly improved. Q1 & Q3] To me it seems that my questions remained largely unanswered here. No example of a function that your model could learn, besides the "not well-defined" functions I discuss in my question, was provided and to me it seems unclear whether such functions exist. In Q4] the part of my question about how models with matching parameter counts perform remains open. Also Q4] remains open in my understanding. Thanks again for your response.

---

> > > ### Author Response · Authors · 2025-11-21
> > >
> > > Thanks for the response.
> > >
> > > 1. The fairest setting is to compare with other node representation methods (e.g., Node2Vec), Node2Net does show significant performance improvement. Like discussed in the paper, many DNN models are developed to enlarge and enrich the design space and toolbox of DNN field, which is sufficient as long as performance is consistent with existing methods. For example, as shown in Table 2, the influential GATV2 published in ICLR 2022 (cited over 2000 times) underperforms about 2% below the 2017 GCN as shown in Table 2 with 100 runs. But its introduction of attention mechanism (along with the GAT paper) provides a new design option to GNN, just like our Node2Net approach.
> > >
> > > 2. These "not well-defined" functions do exist, such as the simple Xor function discussed in page 2.

---

### Official Review · Reviewer_rnaq · 2025-11-01

**Soundness:** 2
**Presentation:** 2
**Contribution:** 1
**Rating:** 2
**Confidence:** 3

**Summary:**

This paper introduces Node2Net, a node-specific parameterization method designed to enhance the expressiveness of GNNs. The authors equip each node with a learnable function to model nonlinear feature interactions while preserving feature-dependent variability. This approach is akin to adding node IDs to each node, where an MLP is learned for each node in a manner similar to Node2Vec. Node2Net extends the representational power of GNNs beyond 1-WL indistinguishability while maintaining linear scaling in both computation and memory. Experiments are conducted to validate the proposed approach.

**Strengths:**

- The writing is adequate.
- The idea of learning node embeddings with separate MLPs is novel, though it has not yet been fully developed into a functional approach that can be truly useful.

**Weaknesses:**

- Why are Node MLPs initially trained as identity mappings?
- There is no clear advantage demonstrated in the GNN comparison experiments.
- The authors claim that Node2Net does not alter the transductive or inductive generalization properties of the backbone. However, the inductive part of this claim is not convincing.
- The authors do not claim permutation invariance in expectation. Several works, such as PF-GNN, perform inductive modeling with learnable positional encodings. Given that both approaches aim to break the symmetry of graphs with learnable position embeddings, the authors should compare their method with PF-GNN.

**Questions:**

Please see weaknesses

---

> ### Author Response · Authors · 2025-11-14
> **Response to the four weaknesses**
>
> 1. Due to a dataset splitting to training, validation, and testing sets, some nodes appearing in testing set do not appear in training set. Pretraining of all nodes with identity mapping makes sure nodes only appearing in test set will input and also output the original features since they will not be trained during training phase of Node2Net. Of course, one alternative is to apply Node2Net only to nodes appearing in training set.
>
> 2. The performance in the GNN field has been stalled in the last 7-8 years actually since GCN in 2017. Take Cora dataset as an example, according to the Cora leaderboard https://hyper.ai/en/sota/tasks/node-classification/benchmark/node-classification-on-cora, it seems that recent methods made a significant improvement on performance (around 90% now over GCN's ~81%). However these recent methods often adopted a different split (70%-80% of 2708 nodes in training set vs. 140 out of 2708 nodes (5%) in training in original GCN setting). Using the 5% original split in GCN paper, probably none of GNN papers in the last 7-8 years can beat the GCN performance (orignal GCN paper reported ~81% Cora accuracy, but based on our experiments, GCN's performance is around ~83%). All of our experiments are run 100 times to make sure our improvement is real and not due to randomness in training. Loss is an important measure for machine learning methods since it usually directly optimized. Our experiments on losses do show significant improvement over baselines.
>
> The most fair baseline to compare with is existing node representation methods, and Node2Net shows significant improvement over the classic Node2Vec.
>
> The contributions of our Node2Net approach are two-fold: theoretically we propose a more reasonable way to id a node and breaks 1-WL indistinguishability than current NodeID methods; at the model architecture level, Node2Net expands the design space of GNN proposed in https://proceedings.neurips.cc/paper/2020/file/c5c3d4fe6b2cc463c7d7ecba17cc9de7-Paper.pdf. Performance-wsie, the improvement is small but consistent across multiple benchmarks whose performance has been stalled for a long time. The small improvement is due to the data split, for example, in Cora only 140 out of 2708 nodes (5%) are trained with Node2Net. In node-level tasks, due to the constraint of data split (nodes in testing set can not be trained in training set), node representation methods including Node2Net will better fit in subgraph-level and graph-level tasks, which we will explore next. In this work, we still want to show Node2Net's effectiveness on the fundamental node level, even if it is not the optimal setting for node representation methods.
>
> 3. Node2Net operates at the level of nodes and do not make any changes to the levels above (e.g., aggregation, pooling). Hence, if a GNN method is inductive/transductive, it will remain the same way.
>
> 4. Thanks for the advice. Due to the computation constraint, we are not able to finish the experiment applying Node2Net in PF-GNN during the rebuttal period. We will include it in the revised version.

---

### Meta-Review · Area_Chair_jDu7 · 2026-01-13

**Summary:**

Across all four reviews, the dominant view is that Node2Net is currently an underdeveloped idea. Its empirical benefits appear insufficient relative to the additional complexity introduced, not commensurate with its additional complexity, and the paper does not provide convincing evidence to substantiate its claims on generalization and scalability. This consensus led to a reject recommendation **(all four reviewers rated the paper “2: reject, not good enough”)**.

**Reviewer Concerns:**

## Concerns discussed/addressed (partially) by the rebuttal:

1. **Why initialize node MLPs as identity mappings?** Authors provided a concrete rationale tied to dataset splits: “some nodes appearing in testing set do not appear in training set” and identity pretraining ensures unseen test nodes “will input and also output the original features” (Authors’ response to Reviewer rnaq).

2. **Disclosure of LLM usage and presentation style**. Reviewer abjw noted lists “reminiscent of the writing style of an LLM … not disclosed.” Authors explicitly acknowledged: “we used LLM to fix grammar and writing” and committed to reformat in the revision (Authors’ response to Reviewer abjw).

3. **Rough estimates of overhead**. In response to requests for costs, authors provided back-of-the-envelope parameter counts and runtime expectations: for Cora, “\~31M” added parameters for all nodes, “\~1.6M” if only training nodes, and “Usually the running time roughly doubles comparing with base model” (Authors’ response to Reviewer abjw; similarly to Reviewer WvU1 and gMZx). They also suggested mitigation via “apply Node2Net only to nodes appearing in training set” and “lower the number of input features” / low-rank embeddings (Authors’ responses).

4. **Some conceptual clarification on expressiveness**. Authors asserted: “A shared MLP will not break 1-WL indistinguishability” (Authors’ response to Reviewer abjw) and later pointed to an existence example (“such as the simple Xor function discussed in page 2”) (Authors’ follow-up to Reviewer abjw).

## Outstanding concerns (not resolved by rebuttal):

1. **Empirical benefit remains unconvincing relative to claimed overhead**. Reviewer abjw concluded post-rebuttal: “manuscript is not ready for publication yet … maintain my score,” emphasizing “no solid argument for roughly doubling the runtime … when … performance is not significantly improved” (Reviewer abjw). Reviewer gMZx similarly stated: “it does not adequately address my concerns … Cora dataset is not convincing … I will maintain my original score” (Reviewer gMZx). The rebuttal did not introduce new experiments that materially change the reported effect sizes.

2. **Fairness controls and “what can Node2Net learn that shared MLP cannot?” remain inadequately answered**. Reviewer abjw stated: “Q1 & Q3 … remained largely unanswered … No example … besides the ‘not well-defined’ functions … unclear whether such functions exist,” and flagged that “matching parameter counts … remains open” (Reviewer abjw, post-rebuttal comment). Authors’ reply (XOR) may not address the reviewer’s specific request for a function class separation beyond ill-defined mappings, nor the parameter-matched baselines.

3. **Inductive/generalization claims remain weak**. Although authors argue node-representation methods are constrained by splits and suggest Node2Net “better fit for subgraph-level and graph-level tasks” (Authors’ responses), the paper as reviewed still lacks evidence that the inductive claims are valid in realistic unseen-node settings (Reviewer rnaq: “inductive … not convincing”; Reviewer WvU1: “pre-training … problematic”).

4. **Scalability to large graphs remains unvalidated**. Reviewers requested either theory or experiments at much larger scale; gMZx explicitly asked for “theoretical analyses … and experimental results on graphs containing tens of millions of nodes” and found the rebuttal insufficient (Reviewer gMZx). Authors largely provided estimates and future-work statements (“we will include … in the revised version,” “we will expand … next step”) rather than concrete scaling evidence.

5. **Missing/insufficient baselines and scope (Graph Transformers)**. Requested comparisons (e.g., PF-GNN) were not completed: “Due to the computation constraint, we are not able to finish … PF-GNN … during the rebuttal period” (Authors’ response to Reviewer rnaq). Reviewer abjw’s concern that the transformer variant is evaluated on too few datasets (“only measured on one dataset”) also remains open, with authors again citing compute constraints and deferring to a revised version.

**Reviewer Scores:**

- **Reviewer rnaq (Rating: 2): No score update was posted in discussion**. Given that key concerns remain unaddressed in the rebuttal (notably “no clear advantage … in the GNN comparison experiments,” and the request to compare with PF-GNN, which authors deferred: “not able to finish … PF-GNN during the rebuttal period”), I expect the reviewer would keep the score at 2 (Reviewer rnaq; Authors’ response to Reviewer rnaq).

- **Reviewer abjw (Rating: 2): Unchanged**. The reviewer explicitly states: “I choose to maintain my score therefore” (Reviewer abjw, post-rebuttal comment).

- **Reviewer WvU1 (Rating: 2): No discussion update was posted**. While the rebuttal provides rough parameter/runtime estimates (e.g., “running time roughly doubles”), it does not add the concrete overhead measurements, large-scale validation, or stronger empirical improvements that this review emphasized (“runtime and memory overhead … is not reported,” “scalability … unclear,” “pre-training … problematic”). I therefore expect the score would remain 2 (Reviewer WvU1; Authors’ response to Reviewer WvU1).

- **Reviewer gMZx (Rating: 2): Unchanged**. The reviewer explicitly concludes: “it does not adequately address my concerns … Therefore, I will maintain my original score” (Reviewer gMZx, post-rebuttal comment).

---

### Decision · Program_Chairs · 2026-01-26

Reject